# ANY-STEP GENERATION VIA $N$-TH ORDER RECURSIVE CONSISTENT VELOCITY FIELD ESTIMATION

**Peng Sun**[1,2]  **Tao Lin**[2,*]
[1]Zhejiang University  [2]Westlake University
sunpeng@westlake.edu.cn, lintao@westlake.edu.cn

## ABSTRACT

Recent advances in few-step generative models (typically 1-8 steps), such as consistency models, have yielded impressive performance. However, their broader adoption is hindered by significant challenges, including substantial computational overhead, the reliance on complex multi-component loss functions, and intricate multi-stage training strategies that lack end-to-end simplicity. These limitations impede their scalability and stability, especially when applied to large-scale models. To address these issues, we introduce $N$-th order Recursive Consistent velocity field estimation for Generative Modeling (RCGM), a novel framework that unifies many existing approaches. Within this framework, we reveal that conventional one-step methods, such as consistency and MeanFlow models, are special cases of 1st-order RCGM. This insight enables a natural extension to higher-order scenarios ($N \geq 2$), which exhibit markedly improved training stability and achieve state-of-the-art (SOTA) performance. For instance, on ImageNet $256 \times 256$, RCGM enables a 675M parameter diffusion transformer to achieve a $1.48$ FID score in just 2 sampling steps. Crucially, RCGM facilitates the stable full-parameter training of a large-scale (20B) unified multi-modal model, attaining a $0.86$ GenEval score in $4$ steps. In contrast, conventional 1st-order approaches, such as consistency and MeanFlow models, typically suffer from training instability, model collapse, or memory constraints under comparable settings. Code is available at: https://github.com/LINs-lab/RCGM.

## 1 INTRODUCTION

Existing PF-ODE-based generative models (Song et al., 2020b), encompassing diffusion models (Ho et al., 2020; Song et al., 2020a), flow-matching models (Lipman et al., 2022; Ma et al., 2024), and consistency models (Song et al., 2023; Lu & Song, 2024), have demonstrated remarkable success in synthesizing high-fidelity data across diverse applications, including image and video generation (Google, 2025a; OpenAI, 2025; Xie et al., 2024a; Ho et al., 2022; Chen et al., 2025c; Wu et al., 2025a).

Table 1: Comparison of different methods' **reliance** on a 1st-order objective and JVP. Our method is independent of both.

| Method | Beyond | |
|---|---|---|
| | 1st-Order | JVP |
| CM (Song et al., 2023) | ✗ | ✓ |
| sCM (Lu & Song, 2024) | ✗ | ✗ |
| MeanFlow (Geng et al., 2025) | ✗ | ✗ |
| RCGM (Ours) | ✓ | ✓ |

Within this landscape, few-step generative models (Song et al., 2023; Frans et al., 2024; Geng et al., 2025) are particularly prized for their ability to generate high-quality samples with significantly reduced computational cost, a critical factor for practical deployment. However, the pursuit of this efficiency has introduced a distinct set of formidable challenges that plague current SOTA methods: **(a)** a prohibitive computational and memory burden during training, often necessitating expensive Jacobian-vector products (JVP) (Geng et al., 2025; Lu & Song, 2024); **(b)** the need to combine multiple losses and train auxiliary models, e.g., combining consistency loss with adversarial loss (Chen et al., 2025c) or training an additional fake image generation model (Yin et al., 2024b;a; Sauer et al., 2024a); **(c)** a fractured theoretical landscape, where highly related methods like consistency models (Song et al., 2023), shortcut models (Frans et al., 2024), and MeanFlow (Geng et al., 2025) have been developed in isolation, lacking a common theoretical foundation.

---

*Corresponding author.

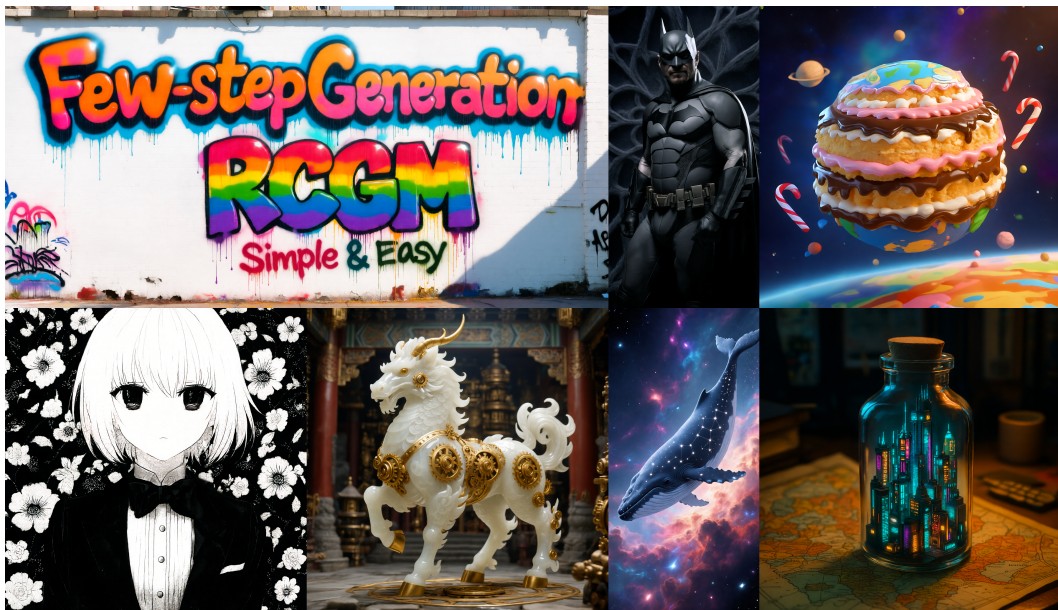

Figure 1: **Visualization results of RCGM on Qwen-Image-20B.** The images shown were generated by the RCGM-tuned Qwen-Image-20B model using **NFE=8** (GenEval score=0.87). Please zoom in to see finer details.

These challenges restrict their broader application, particularly in generalizing to large-scale models with guaranteed stability and efficiency. For instance, our experiments show that existing one-step models, such as consistency models, often suffer from training instability and high computational demands when scaled up, frequently resulting in model collapse or GPU memory exhaustion (see Tab. 4). We argue that this fragility fundamentally stems from their implicit reliance on a 1st-order recursive training objective (cf. Sec. 2 and Sec. 3). Crucially, while these few-step models rely heavily on self-targeted learning, their 1st-order recursive nature is inherently incompatible with Exponential Moving Average (EMA) techniques. Consequently, they are deprived of the stable training targets and smoothed optimization dynamics that EMA typically provides, creating a profound bottleneck for scaling. This critical insight leads us to the central question:

> **Problem 1 .** *Can we develop a unified and simple framework that: (a) encompasses existing few-step generative models as a special case; (b) enhances training stability and generalization to large-scale models by moving beyond the 1st-order limitation—thereby unlocking compatibility with stabilizing techniques like EMA and obviating the need for JVP or training auxiliary models?*

To address these challenges, we propose RCGM, a novel and principled framework that unifies and generalizes existing approaches. *Within our framework, we show that conventional one-step models (e.g., consistency models and MeanFlow) correspond to the special case of 1st-order* RCGM.

Notably, RCGM naturally supports higher-order formulations (i.e., $N \geq 2$). These higher-order variants utilize more comprehensive trajectory information from the PF-ODE, which contributes to substantially improved training stability. This stability enables successful training in demanding large-scale settings where 1st-order models often fail, ultimately achieving SOTA performance without resorting to complex workarounds. **In summary, our contributions are:**

(a) We propose RCGM, a unified framework that contextualizes existing few-step generative models as a specific 1st-order case and generalizes them to arbitrary $N$-th order formulations.

(b) We identify the *EMA incompatibility paradox* inherent in 1st-order self-targeted learning. We empirically verify that higher-order RCGM (e.g., 2nd-order) elegantly breaks this paradox, successfully harnessing the stabilizing benefits of aggressive EMA smoothing to exhibit superior robustness and enabling effective scaling to larger, more complex model architectures (cf. Sec. 4).

(c) Our method achieves SOTA performance across a range of standard benchmarks, outperforming existing methods in few-step generation tasks while maintaining computational efficiency.

Supported by Fig. 1, Sec. 4, and App. D, our approach matches or outperforms SOTA methods across diverse datasets, architectures, and resolutions, advancing efficient, high-fidelity generation.

## 2 Preliminaries

Let $p(\mathbf{x})$ be the data distribution for a given training set $D$. This distribution can also be conditional, denoted as $p(\mathbf{x}|\mathbf{c})$ for a given condition $\mathbf{c}$. Diffusion-based generative models aim to learn a transformation from a simple prior distribution $p(\mathbf{z})$, typically the standard Gaussian $\mathcal{N}(\mathbf{0}, \mathbf{I})$, to the complex target data distribution $p(\mathbf{x})$.

This is often achieved by learning to reverse a forward noising process. The forward process gradually perturbs a clean data sample $\mathbf{x} \sim p(\mathbf{x})$ into a noisy intermediate sample $\mathbf{x}_t$ using a predefined trajectory, such as $\mathbf{x}_t = \alpha(t)\mathbf{z} + \gamma(t)\mathbf{x}$, where $\mathbf{z} \sim \mathcal{N}(\mathbf{0}, \mathbf{I})$. The time variable $t$ spans the interval $[0, 1]$, with the perturbation effect intensifying as $t$ increases. The scheduling functions $\alpha(t)$ and $\gamma(t)$ are continuously differentiable, i.e., $\alpha(t), \gamma(t) \in C^1[0, 1]$, and satisfy the boundary conditions: $\alpha(0) = 0, \gamma(0) = 1$ (yielding the data) and $\alpha(1) = 1, \gamma(1) = 0$ (yielding pure noise).

More formally, diffusion models learn a function that guides the transformation of samples along the trajectory of the Probability Flow Ordinary Differential Equation (PF-ODE) (Song et al., 2020b), which connects the prior distribution $p(\mathbf{z})$ to the data distribution $p(\mathbf{x})$.

In this paper, we define a general prediction function $-\boldsymbol{f}(\mathbf{x}_t, r) := \mathbf{x}_r - \mathbf{x}_t$ that estimates the displacement from $\mathbf{x}_t$ to a target $\mathbf{x}_r$, with further details in (6). This function aims to predict the target point $\mathbf{x}_r$ from the current point $\mathbf{x}_t$ along a specific PF-ODE trajectory. In the following sections, we will introduce several prominent learning paradigms for deep generative models.

### 2.1 0th-order: Diffusion and Flow-Matching Models

**Diffusion and Flow-Matching Models (Ho et al., 2020; Song et al., 2020b; Lipman et al., 2022; Sun et al., 2025).** Recent work by Sun et al. (Sun et al., 2025) established a unified framework for diffusion and flow-matching models. This framework reveals that both paradigms aim to learn the same PF-ODE (1), but they differ in their underlying transport processes (i.e., their specific choices of $\alpha(t)$ and $\gamma(t)$) and training objectives.

Specifically, a neural network $\boldsymbol{F}_{\boldsymbol{\theta}}$ is trained by minimizing a general objective of the form: $\mathbb{E}_{\mathbf{x}_t, t}[d(\boldsymbol{F}_{\boldsymbol{\theta}}(\mathbf{x}_t), \hat{\alpha}(t)\mathbf{z} + \hat{\gamma}(t)\mathbf{x})]$, where $d(\cdot, \cdot)$ denotes a distance metric. As derived in (Sun et al., 2025), the output of the trained network, $\boldsymbol{F}_t := \boldsymbol{F}_{\boldsymbol{\theta}}(\mathbf{x}_t)$, can be used to construct the component functions: $\boldsymbol{f}^{\mathbf{x}}(\boldsymbol{F}_t, \mathbf{x}_t, t) := \frac{\alpha(t) \cdot \boldsymbol{F}_t - \hat{\alpha}(t) \cdot \mathbf{x}_t}{\alpha(t) \cdot \hat{\gamma}(t) - \hat{\alpha}(t) \cdot \gamma(t)}$ and $\boldsymbol{f}^{\mathbf{z}}(\boldsymbol{F}_t, \mathbf{x}_t, t) := \frac{\hat{\gamma}(t) \cdot \mathbf{x}_t - \gamma(t) \cdot \boldsymbol{F}_t}{\alpha(t) \cdot \hat{\gamma}(t) - \hat{\alpha}(t) \cdot \gamma(t)}$. These components, in turn, define the velocity field of the PF-ODE: $\frac{\mathrm{d}\mathbf{x}_t}{\mathrm{d}t} = \frac{\mathrm{d}\alpha(t)}{\mathrm{d}t} \cdot \boldsymbol{f}^{\mathbf{z}}(\boldsymbol{F}_t, \mathbf{x}_t, t) + \frac{\mathrm{d}\gamma(t)}{\mathrm{d}t} \cdot \boldsymbol{f}^{\mathbf{x}}(\boldsymbol{F}_t, \mathbf{x}_t, t)$. The sampling process then involves numerically integrating this velocity field to solve the PF-ODE. The integration proceeds backward in time, starting from a prior sample $\mathbf{x}_1 \sim p(\mathbf{z})$ at $t = 1$ and ending at $t = 0$ to produce a data sample from $p(\mathbf{x})$.

Within our framework, we adopt a zeroth-order inductive learning perspective to interpret this process, a view supported by Fig. 3 (a). Specifically, for a sufficiently small step $\Delta t$, the prediction function's learning target becomes the product of the velocity field and the time step:

$$\boldsymbol{f}(\mathbf{x}_t, t - \Delta t) \leftarrow \frac{\mathrm{d}\mathbf{x}_t}{\mathrm{d}t} \cdot \Delta t \quad \text{as} \quad \Delta t \rightarrow 0.$$

In essence, given the current state $\mathbf{x}_t$, the prediction function $\boldsymbol{f}$ directly learns to predict the displacement required to approximate the next state, $\mathbf{x}_{t-\Delta t}$, on the PF-ODE path.

### 2.2 1st-order: Recursive Consistency Models

**Consistency Models (Song et al., 2023; Lu & Song, 2024; Sun et al., 2025).** Consistency models are designed to bypass the iterative nature of diffusion models. Their primary goal is to learn a function that maps any noisy state $\mathbf{x}_t$ directly to the clean data endpoint $\mathbf{x}_0$ in a single step. This is achieved by estimating the endpoint of the PF-ODE trajectory originating from $\mathbf{x}_t$, using the function $\mathbf{x}_0 = \boldsymbol{f}^{\mathbf{x}}(\boldsymbol{F}_t, \mathbf{x}_t, t)$.

The training objective is specifically designed to instill a crucial "consistency" property. This property ensures coherence between the model's predictions for the clean data, even when originating from two temporally adjacent noisy states that are separated by a finite time interval $\Delta t > 0$:

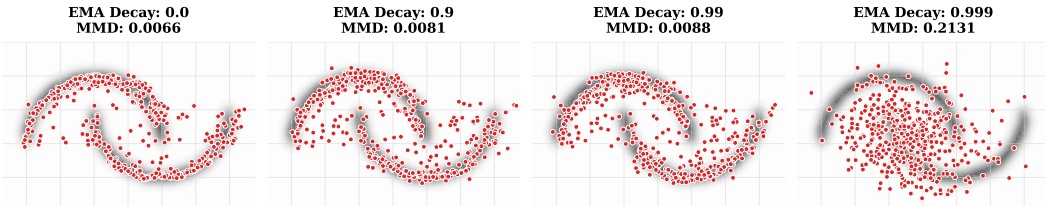

Figure 2: **The brittleness of 1st-order models under aggressive EMA decay rate ($\kappa$).** We evaluate a representative consistency-based model (Song et al., 2023) on the Moons dataset. Gray areas: Ground truth; Red dots: One-step samples. We adopt Maximum Mean Discrepancy (MMD) as a non-parametric metric to measure the divergence between the generated and real distributions. As $\kappa$ approaches unity (e.g., 0.999), we observe a significant degradation in sample quality (indicated by a higher MMD and poor visual fidelity), suggesting a fundamental incompatibility between 1st-order recursion and aggressive temporal smoothing.

$\mathbb{E}_{\mathbf{x}_t,t}\left[d\left(\boldsymbol{f}^{\mathbf{x}}(\boldsymbol{F}_t,\mathbf{x}_t,t),\mathrm{stopgrad}(\boldsymbol{f}^{\mathbf{x}}(\boldsymbol{F}_{t-\Delta t},\mathbf{x}_{t-\Delta t},t-\Delta t)))\right]$. A known limitation of discrete-time consistency models is their sensitivity to the choice of $\Delta t$, which often requires manually tuned annealing schedules for efficient training (Song & Dhariwal, 2023; Geng et al., 2024). This challenge was later addressed by continuous consistency models, which derive their training objective by taking the limit as $\Delta t \to 0$ (Lu & Song, 2024).

We interpret this process through a **1st-order inductive learning** lens, a perspective supported by our visualizations in Fig. 3 and theoretical analysis in App. E.1. This view frames the learning objective as a recursive formulation:

$$\boldsymbol{f}(\mathbf{x}_t, 0) \leftarrow \frac{\mathrm{d}\mathbf{x}_t}{\mathrm{d}t} \cdot \Delta t + \boldsymbol{f}(\mathbf{x}_{t-\Delta t}, 0)$$

This recursive principle—approximating a long-range prediction by combining an infinitesimal step with another long-range prediction—is also reflected in follow-up works (Frans et al., 2024; Geng et al., 2025). For instance, shortcut models (Frans et al., 2024) employ a similar self-recursive formulation and generalize it to predict between arbitrary time points $t$ and $r \in [0, t]$: $\boldsymbol{f}(\mathbf{x}_t, r) \leftarrow \boldsymbol{f}(\mathbf{x}_t, s) + \boldsymbol{f}(\mathbf{x}_s, r)$. This is then combined with a flow-matching objective to train one-step generative models. More recently, MeanFlow (Geng et al., 2025) extended this idea by training a one-step model with the recursive objective $\boldsymbol{f}(\mathbf{x}_t, r) \leftarrow \frac{\mathrm{d}\mathbf{x}_t}{\mathrm{d}t} \cdot \Delta t + \boldsymbol{f}(\mathbf{x}_{t-\Delta t}, r)$ for any target time $r$.

In summary, while diffusion and flow-matching models are inherently multi-step frameworks, consistency models represent a paradigm shift towards few-step or one-step generation.

**The Exponential Moving Average (EMA) incompatibility paradox.** Standard practice in self-targeted learning (Grill et al., 2020; He et al., 2020; Caron et al., 2021; Song et al., 2023) involves maintaining an EMA version of the model, $\boldsymbol{f}_{\boldsymbol{\theta}^-}$, to provide stable regression targets for the online model $\boldsymbol{f}_{\boldsymbol{\theta}}$. Intuitively, one would expect high EMA decay rates (e.g., $\kappa = 0.999$) to enhance training stability by filtering out optimization noise.

However, we identify a fundamental incompatibility within existing 1st-order consistency-based models: the recursive formulation inherent to 1st-order objectives actively conflicts with the delayed updates of an EMA model, resulting in substantial performance degradation at high decay rates.

To systematically investigate this phenomenon, Fig. 2 details an empirical analysis of how varying the EMA decay rate alters the training dynamics of 1st-order models, with large-scale evaluations provided in Sec. 4.2. Our findings reveal an acute sensitivity to the decay hyperparameter. Specifically, aggressive decay rates (e.g., 0.999) severely compromise the fidelity of one-step generation, whereas relaxed rates (e.g., 0.9) fail to yield meaningful improvements over naive, non-EMA self-targeting.

## 3 METHODOLOGY

We begin by deriving a recursive, $N$-th order velocity field estimator through the segmented integration of the Probability Flow ODE (PF-ODE) trajectory (Sec. 3.1). Building on this formulation, we introduce a unified training objective that enables any-step generation (Sec. 3.2). Finally, we discuss key practical considerations for implementing our method, RCGM (Sec. 3.3).

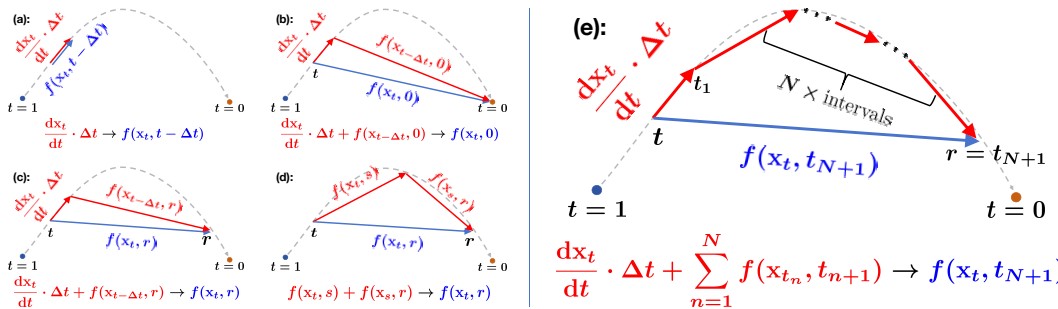

Figure 3: **A conceptual illustration of our proposed framework, RCGM, which generalizes existing generative models by formulating them within a unified higher-order structure. Trajectories map a current state to a target learning state.** **(a)** Standard diffusion (Ho et al., 2020) and flow-matching (Lipman et al., 2022) models correspond to the 0th-order case ($N = 0$) of our framework. **(b-d)** Prominent one-step models, including consistency models (Song et al., 2023), MeanFlow (Geng et al., 2025), and shortcut models (Frans et al., 2024), are special instances of the 1st-order case ($N = 1$). **(e)** RCGM extends this hierarchy to arbitrary orders ($N \geq 0$), enabling the use of higher-order information for potentially more robust training dynamics.

### 3.1 SEGMENTED INTEGRATION ALONG THE PF-ODE TRAJECTORY

Our methodology is grounded in the PF-ODE formulation, where a trajectory from a prior distribution to the data distribution is defined by a velocity field $\mathbf{v}(\mathbf{x}_\tau, \tau)$. For a diffusion process specified by $\mathbf{x}_t = \alpha(t)\mathbf{z} + \gamma(t)\mathbf{x}_0$, this velocity is given by (Song et al., 2020b; Sun et al., 2025):

$$\mathbf{v}(\mathbf{x}_\tau, \tau) := \frac{\gamma'(\tau)}{\gamma(\tau)}\mathbf{x}_\tau - \left[\alpha(\tau)\alpha'(\tau) - \frac{\gamma'(\tau)}{\gamma(\tau)}\alpha(\tau)^2\right]\nabla_{\mathbf{x}_\tau}\log p_\tau(\mathbf{x}_\tau). \quad (1)$$

The integral form of this ODE connects any two points $\mathbf{x}_t$ and $\mathbf{x}_{t_{N+1}}$ on a trajectory. We proceed by partitioning the integration interval $[t, t_{N+1}]$ with $N$ intermediate points, where $t = t_0 > t_1 > \cdots > t_{N+1}$. This segmentation allows us to decompose the total displacement into a sum over the sub-intervals:

$$\mathbf{x}_{t_{N+1}} - \mathbf{x}_t = \sum_{i=0}^{N}\int_{t_i}^{t_{i+1}}\mathbf{v}(\mathbf{x}_\tau, \tau)\mathrm{d}\tau = \int_{t_0}^{t_1}\mathbf{v}(\mathbf{x}_\tau, \tau)\mathrm{d}\tau + \sum_{i=1}^{N}\int_{t_i}^{t_{i+1}}\mathbf{v}(\mathbf{x}_\tau, \tau)\mathrm{d}\tau. \quad (2)$$

The core of our approach is to approximate the first integral segment. For a sufficiently small time step $\Delta t := t_0 - t_1$, a 1st-order Taylor approximation (i.e., a forward Euler step) is justified:

$$\int_{t_0}^{t_1}\mathbf{v}(\mathbf{x}_\tau, \tau)\mathrm{d}\tau \approx -\mathbf{v}(\mathbf{x}_{t_0}, t_0)\Delta t = -\frac{\mathrm{d}\mathbf{x}_t}{\mathrm{d}t}\Delta t. \quad (3)$$

Substituting this approximation into the exact identity from (2) yields the relationship:

$$\mathbf{x}_{t_{N+1}} \approx \mathbf{x}_t - \frac{\mathrm{d}\mathbf{x}_t}{\mathrm{d}t}\Delta t + \sum_{i=1}^{N}\int_{t_i}^{t_{i+1}}\mathbf{v}(\mathbf{x}_\tau, \tau)\mathrm{d}\tau. \quad (4)$$

By rearranging (4), we obtain our final estimator. We define the **recursive $N$-th order velocity field estimation** as the target derived from this multi-step formula:

$$\frac{\mathrm{d}\mathbf{x}_t}{\mathrm{d}t} \approx -\frac{1}{\Delta t}\left[(\mathbf{x}_{t_{N+1}} - \mathbf{x}_t) - \sum_{i=1}^{N}\int_{t_i}^{t_{i+1}}\mathbf{v}(\mathbf{x}_\tau, \tau)\mathrm{d}\tau\right]. \quad (5)$$

This formulation is termed **recursive** because the estimation of the velocity $\mathbf{v}$ at time $t$ depends on the integral of the same velocity field over future time steps. The "$N$-th order" designation refers to the $N$ integral correction terms that refine the estimate beyond a simple one-step approximation, thereby providing a more accurate target for model training.

### 3.2 A UNIFIED TRAINING FRAMEWORK FOR ANY-STEP GENERATION

Our goal is to train an *any-step* generative model capable of predicting the state $\mathbf{x}_r$ at any future time $r < t$ from the current state $\mathbf{x}_t$ along a given PF-ODE trajectory. To this end, we define a

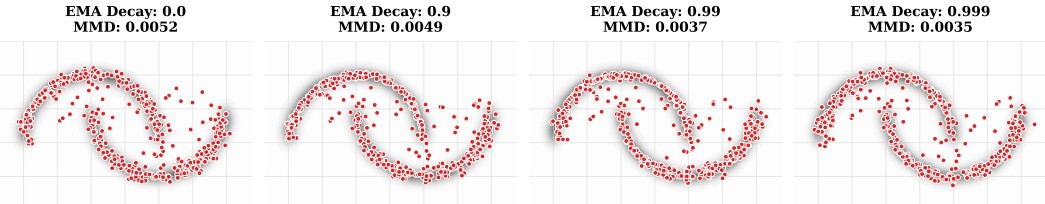

Figure 4: **Robustness of 2nd-order models across varying EMA decay rates ($\kappa$).** We evaluate RCGM configured with $N = 2$ under identical conditions to Fig. 2. Even when employing the aggressive EMA decay rates (e.g., $\kappa = 0.999$) that caused significant degradation in 1st-order models, we observe that sample quality remains consistently high across all configurations. This is evidenced by low MMD values and visually faithful generations. This empirical evidence strongly supports our hypothesis that higher-order formulations enable effective training under aggressive temporal smoothing in self-targeted learning, thereby mitigating the associated instability issues.

*displacement function* $\boldsymbol{f}(\mathbf{x}_t, r)$ that maps the current state to the total displacement required to reach the target state:

$$-\boldsymbol{f}(\mathbf{x}_t, r) := \mathbf{x}_r - \mathbf{x}_t = \int_t^r \mathbf{v}(\mathbf{x}_\tau, \tau)\mathrm{d}\tau, \quad r \in [0, t]. \tag{6}$$

Using this definition, we can reformulate the recursive $N$-th order velocity estimator from (5) entirely in terms of displacements:

$$\frac{\mathrm{d}\mathbf{x}_t}{\mathrm{d}t} \approx \frac{1}{\Delta t}\left[\boldsymbol{f}(\mathbf{x}_t, t_{N+1}) - \sum_{i=1}^{N} \boldsymbol{f}(\mathbf{x}_{t_i}, t_{i+1})\right]. \tag{7}$$

This identity forms the foundation of our training objective. It provides a multi-step target for the instantaneous velocity $\mathrm{d}\mathbf{x}_t/\mathrm{d}t$, which is known analytically from the PF-ODE formulation (cf. (1)).

We parameterize the displacement function with a parameterized model $\boldsymbol{f}_{\boldsymbol{\theta}}(\mathbf{x}_t, r)$. To train $\boldsymbol{f}_{\boldsymbol{\theta}}$, we enforce the identity in (7). The terms $\boldsymbol{f}(\mathbf{x}_{t_i}, t_{i+1})$ for $i \geq 1$ represent future displacements and are treated as fixed targets during optimization. Following standard practice in consistency training (Song et al., 2023), we use a target model $\boldsymbol{f}_{\boldsymbol{\theta}^-}$ (e.g., an exponential moving average of $\boldsymbol{\theta}$ or a periodically updated copy) for these terms, applying a stop-gradient operator to prevent backpropagation through them. This yields the following learning objective:

$$\mathcal{L}(\boldsymbol{\theta}) = \mathbb{E}_{\mathbf{x}_0, \mathbf{z}, \{t_i\}_{i=0}^{N+1}}\left[d\left(\underbrace{\frac{\mathrm{d}\mathbf{x}_t}{\mathrm{d}t}}_{\text{True Velocity}}, \underbrace{\frac{1}{\Delta t}\left[\boldsymbol{f}_{\boldsymbol{\theta}}(\mathbf{x}_t, t_{N+1}) - \sum_{i=1}^{N} \boldsymbol{f}_{\boldsymbol{\theta}^-}(\mathbf{x}_{t_i}, t_{i+1})\right]}_{\text{Model's Velocity Estimate}}\right)\right], \tag{8}$$

where $\mathbf{x}_t = \alpha(t)\mathbf{z} + \gamma(t)\mathbf{x}_0$ with $\mathbf{z} \sim \mathcal{N}(\mathbf{0}, \mathbf{I})$, time points are sampled hierarchically (e.g., $t \sim \mathcal{U}[0, T]$, $t_1 \sim \mathcal{U}[0, t]$, etc.), and $d(\cdot, \cdot)$ is a suitable metric, such as the squared $\ell_2$-norm.

This unified formulation elegantly generalizes several established generative modeling paradigms:

(a) **For $N = 0$,** the objective simplifies to matching $\mathrm{d}\mathbf{x}_t/\mathrm{d}t$ with $\boldsymbol{f}_{\boldsymbol{\theta}}(\mathbf{x}_t, t_1)/\Delta t$. This is equivalent to the objectives used in score-based diffusion models (Song et al., 2020b) and flow matching (Lipman et al., 2022).

(b) **For $N = 1$,** the objective corresponds to those of one-step consistency models (Song et al., 2023; Lu & Song, 2024) and shortcut-based methods (Frans et al., 2024), which use a single future segment to define the training target.

By extending this framework to higher orders ($N \geq 2$), our approach leverages multiple future steps to construct a more robust and stable training signal. As we demonstrate in our experiments (see Fig. 4 and Sec. 4), this generalization improves model performance and convergence across diverse generation tasks.

Notably, regardless of the setting of $N$, our training objective requires only a single model forward pass with gradient calculation and $N$ forward passes without. This design avoids increased GPU

memory costs during training, making it feasible for large-scale models [1]. A detailed discussion on the setting of $N$ is provided in Sec. 4.2.

## 3.3 PRACTICAL IMPLEMENTATION OF RCGM

In this section, we detail several key aspects of the practical implementation of our method, RCGM. We discuss the parameterization of our neural network under a linear transport path, the use of an enhanced target score function to improve performance, the strategy for conditioning the model on both input and target times, and the formulation of a practical loss function for stable and effective training.

**Linear transport and network parameterization.** We employ the linear transport path common in flow-matching literature (Lipman et al., 2022; Ma et al., 2024; Xie et al., 2024a), defined by coefficients $\alpha(t) = t$ and $\gamma(t) = 1 - t$. This transport corresponds to a constant velocity field, implying that the displacement between any two states $\mathbf{x}_t$ and $\mathbf{x}_r$ is directly proportional to the time difference $t - r$. This property motivates our parameterization of the predictive function $\boldsymbol{f}_{\boldsymbol{\theta}}(\mathbf{x}_t, t, r)$, which estimates the displacement from $\mathbf{x}_t$ to $\mathbf{x}_r$, as: $\boldsymbol{f}_{\boldsymbol{\theta}}(\mathbf{x}_t, t, r) = \boldsymbol{F}_{\boldsymbol{\theta}}(\mathbf{x}_t, t, r) \cdot (t - r)$, where $\boldsymbol{F}_{\boldsymbol{\theta}}$ is a neural network designed to approximate the average displacement $(\mathbf{x}_t - \mathbf{x}_r)/(t - r)$.

**Enhanced target score function.** The performance of continuous generative models can be significantly improved by incorporating guidance during the training or sampling process (Ho & Salimans, 2022; Dhariwal & Nichol, 2021; Karras et al., 2022). This is achieved by modifying the conditional target score function from $\nabla_{\mathbf{x}_t} \log p_t(\mathbf{x}_t)$ (defined in (1)) to an enhanced version: $\nabla_{\mathbf{x}_t} \log \left( p_t(\mathbf{x}_t|\mathbf{c}) \left( p_{t,\theta}(\mathbf{x}_t|\mathbf{c})/p_{t,\theta}(\mathbf{x}_t) \right)^\zeta \right)$, where $\zeta \in (0, 1)$ is the enhancement ratio. We follow the same implementation as previous studies (Frans et al., 2024; Sun et al., 2025).

**Input time conditioning.** Our method learns a continuous-time model, $\boldsymbol{f}_{\boldsymbol{\theta}}(\mathbf{x}_t, r)$, designed to predict the state $\mathbf{x}_r$ at a target time $r$ from an initial state $\mathbf{x}_t$ along the probability flow ODE (PF-ODE) trajectory. To accurately map between arbitrary time points, the model must be effectively conditioned on both the input time $t$ and the target time $r$. Following standard practice (Ho et al., 2020; Frans et al., 2024), we employ a time embedding technique where $t$ and $r$ are embedded into vector representations separately. These embeddings, along with the input $\mathbf{x}_t$, are then fed into the neural network $\boldsymbol{F}_{\boldsymbol{\theta}}$, redefining the model as $\boldsymbol{f}_{\boldsymbol{\theta}}(\mathbf{x}_t, r) = \boldsymbol{F}_{\boldsymbol{\theta}}(\mathbf{x}_t, t, r) \cdot (t - r)$.

**Practical loss design.** The training objective in (8) necessitates a carefully designed loss function. While the $\ell_2$-norm is a standard choice for the metric $d(\cdot, \cdot)$ (Ho et al., 2020; Song et al., 2020a), directly optimizing the original objective is suboptimal. We observe that the magnitude of the model output, $\boldsymbol{f}_{\boldsymbol{\theta}}(\mathbf{x}_t, r) \approx \boldsymbol{F}_{\boldsymbol{\theta}} \cdot (t - r)$, scales linearly with the time interval $(t - r)$. This introduces an implicit, scale-dependent weighting that causes optimization instability, as larger time steps dominate the gradients. To rectify this bias and stabilize training, we employ a variance-reduction technique inspired by Lu & Song (2024). Specifically, by leveraging the gradient identity derived in previous studies (Lu & Song, 2024; Sun et al., 2025) (i.e., $\nabla_\theta \mathbb{E}[\boldsymbol{F}_{\boldsymbol{\theta}}^\top \boldsymbol{y}] = \frac{1}{2} \nabla_\theta \mathbb{E}[\|\boldsymbol{F}_{\boldsymbol{\theta}} - \boldsymbol{F}_{\theta^-} + \boldsymbol{y}\|_2^2]$), we reformulate our objective into a regression form that decouples the gradient scale from the time interval. Substituting this into (8) yields our final training objective:

$$\mathcal{L}(\boldsymbol{\theta}) = \mathbb{E}_{\mathbf{x}, \mathbf{z}, \{t_i\}_{i=0}^{N+1}} \left[ \left\| \left( \boldsymbol{F}_{\boldsymbol{\theta}}(\mathbf{x}_t, t, t_{N+1}) - \boldsymbol{F}_{\boldsymbol{\theta}^-}(\mathbf{x}_t, t, t_{N+1}) + \xi(\mathbf{x}_t, \{t_i\}_{i=0}^{N+1}) \right) \right\|_2^2 \right], \quad (9)$$

where the target item is $\xi(\mathbf{x}_t, \{t_i\}_{i=0}^{N+1}) := \frac{1}{\Delta t} \left[ \boldsymbol{f}_{\boldsymbol{\theta}^-}(\mathbf{x}_t, t_{N+1}) - \sum_{i=1}^{N} \boldsymbol{f}_{\boldsymbol{\theta}^-}(\mathbf{x}_{t_i}, t_{i+1}) \right] - \frac{d\mathbf{x}_t}{dt}$.

## 4 EXPERIMENTS

This section presents the experimental validation of our proposed methodology, denoted as RCGM. We begin by outlining the experimental setup, including datasets, network architectures, and imple-

---

[1]This is a significant advantage over conventional few-step training methods that often rely on Jacobian-vector products (JVP), which can substantially increase GPU memory consumption (Geng et al., 2025; Lu & Song, 2024). Furthermore, the use of JVP can introduce complex technical challenges when integrating with widely-used architectural optimizations like Flash-Attention (Dao et al., 2022).

mentation details. We then present a comprehensive evaluation of RCGM's performance. Theoretically, our approach converges to conventional flow-matching and diffusion models when $N = 0$. Consequently, to rigorously assess the unique contributions of RCGM, our empirical investigation focuses on the regime where $N \geq 1$.

## 4.1 EXPERIMENTAL SETUP

**Datasets.**    Our primary evaluation is conducted on the ImageNet-1K dataset (Deng et al., 2009), utilizing resolutions of $256 \times 256$ and $512 \times 512$. This choice aligns with established benchmarks in recent high-fidelity generative modeling literature (Karras et al., 2024; Song et al., 2023). We adopt the data preprocessing pipeline from ADM (Dhariwal & Nichol, 2021) to ensure consistency and comparability with prior work. All experiments are performed in the latent space of pretrained autoencoders, a standard practice for efficient training of large-scale models. Specifically:

(a) For $256 \times 256$ images, we leverage widely adopted autoencoders, including the SD-VAE (Rombach et al., 2022) and the VA-VAE (Yao et al., 2025).
(b) For $512 \times 512$ images, in addition to the SD-VAE, we employ a DC-AE (Chen et al., 2024c) with a higher compression ratio (*f32c32*) to mitigate computational demands.

**Network architectures.**    We build upon the success of transformer-based architectures for generative modeling. Our core model is a 675M-parameter Diffusion Transformer (DiT) (Peebles & Xie, 2023), a backbone widely employed in SOTA models such as SiT (Ma et al., 2024), Lightening-DiT (Yao et al., 2025), and DDT (Wang et al., 2025a).

**Implementation details.**    Our models are implemented in PyTorch (Paszke, 2019) and trained using the AdamW optimizer (Loshchilov & Hutter, 2017) with $\beta_1 = 0.9$, $\beta_2 = 0.95$, a constant learning rate of $2 \times 10^{-4}$, and a batch size of 1024. Unless otherwise specified, we train all models using a 2nd-order scheme ($N = 2$). The only exception is during the final 40 epochs of training on ImageNet-1K, where we employ central difference consistency training (Sun et al., 2025) (which can be viewed as a special case of 2nd-order RCGM) to achieve further performance gains. For the time distribution during training, we follow the exact configurations used in Sun et al. (2025).

To evaluate the quality of generated samples, we adhere to standard protocols established in the literature (Song et al., 2020b; Ho et al., 2020; Lipman et al., 2022; Brock et al., 2018). Our primary metric is the Fréchet Inception Distance (FID) (Heusel et al., 2017), computed using $50,000$ generated samples (FID-50K) against the training set.

## 4.2 ANALYSIS OF HIGHER-ORDER TRAINING

It is widely known that training few-step models is challenging due to the instability of training (Song et al., 2023; Lu & Song, 2024), especially when using a large model and a large learning rate, etc.

This issue is more severe when training few-step models in real-world applications such as high-resolution text-to-image generation.

Using Exponential Moving Average (EMA) model in (8) is a key technique help stabilize training and improve performance, which also evidenced in previous 1st-order methods (Song et al., 2023). For those without using EMA model, they typically require a careful technical design to stabilize training, e.g., using `JVP` (Lu & Song, 2024) or careful hyperparameter design (Song & Dhariwal, 2023).

In this section, to investigate how the order $N$ in RCGM affects the training stability and performance under different EMA decay rates $\kappa$, we conduct a series of ablation studies on ImageNet-1K $256 \times 256$ using 675M diffusion transformer with SD-VAE.

**A large EMA decay rate $\kappa$ is critical for 1st-order training stability.**    We first investigate the effect of the EMA decay rate $\kappa$ from (8) on the stability and performance of the conventional 1st-order ($N = 1$) model. As illustrated in Fig. 5a, training without EMA ($\kappa = 0$) is highly unstable, causing the FID score to fluctuate and fail to converge. A small decay rate ($\kappa = 0.9$) tempers this instability, leading to a smoother decrease in FID, yet the final performance remains suboptimal (FID of 31.70). Conversely, while large decay rates ($\kappa \in \{0.99, 0.999\}$) effectively stabilize training dynamics, they

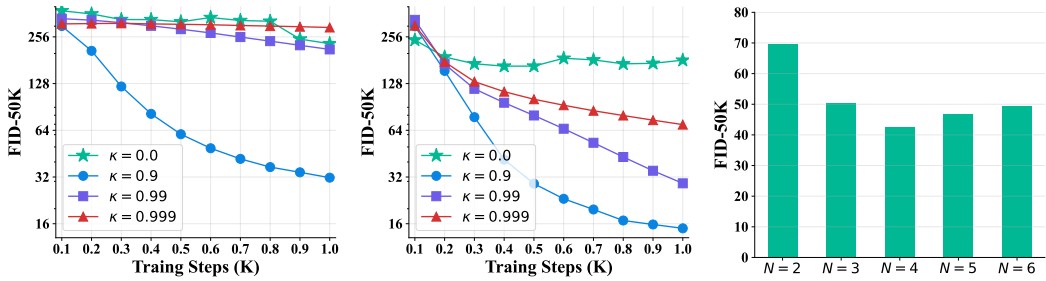

(a) **1st-order over training steps.**   (b) **2nd-order over training steps.**   (c) **Various orders ($\kappa = 0.999$).**

Figure 5: **Ablation studies of RCGM on ImageNet-1K** $256 \times 256$. These studies evaluate key factors of the proposed RCGM for training few-step models, i.e., the order of RCGM ($N$) and the EMA decay rate $\kappa$. The sampling is performed using one step (1-NFE).

severely hinder convergence. This "over-stabilization" is particularly pronounced at $\kappa = 0.999$, where the model converges to a poor FID of 294.18. These results reveal a fundamental tension between training stability and model performance in the 1st-order setting.

**Higher-order approximations resolve the stability-performance trade-off.**   We next examine whether higher-order approximations can alleviate the aforementioned tension. Fig. 5b shows the results for our second-order model ($N = 2$). Strikingly, the second-order model thrives under the large EMA decay rates that were detrimental in the 1st-order case. While the no-EMA ($\kappa = 0$) setting remains unstable and the low-EMA ($\kappa = 0.9$) setting achieves a modest FID of 14.94, the high-EMA regime is transformed. With $N = 2$, a large decay rate such as $\kappa = 0.99$ no longer cripples performance but instead yields a competitive FID of 29.13. This demonstrates that higher-order models possess substantially greater robustness to the choice of $\kappa$, enabling them to benefit from strong EMA stabilization without sacrificing generative quality.

Further experiments, shown in Fig. 5c, confirm that increasing the order $N$ (with a fixed, high $\kappa = 0.999$) can yield additional gains. Performance steadily improves as $N$ increases from 1 to 4, which achieves the lowest FID. However, this trend reverses for $N > 4$, likely due to the accumulation of approximation errors in the higher-order velocity estimates in (8).

*In summary, 1st-order models face a difficult trade-off: a large $\kappa$ is needed for stability but harms final performance. Higher-order methods effectively resolve this conflict, achieving both stable convergence and strong performance with large $\kappa$. Considering the balance between computational cost and performance, we adopt $N = 2$ and $\kappa = 0.999$ as our default configuration.*

### 4.3   COMPARISON WITH SOTA FEW-STEP METHODS

As demonstrated in Tab. 2, our proposed RCGM, when paired with various autoencoders, consistently outperforms or remains highly competitive with SOTA few-step generative models. The following analysis details its advantages across different VAE architectures.

(a) **Performance with SD-VAE** ($256 \times 256$ **and** $512 \times 512$)**:** When paired with a standard SD-VAE, our method exhibits exceptional performance. At $256 \times 256$ resolution, it achieves an FID of 1.92 with 2 NFEs, outperforming IMM's best result while requiring 8 times fewer sampling steps. At $512 \times 512$ resolution, our model achieves an FID of 2.25 with 2 NFEs, which is highly competitive with specialized distillation models like sCD-L (2.04 FID) and sCD-M (2.26 FID), despite their significantly higher training costs (1434 and 1997 epochs vs. our 360).

(b) **Performance with DC-AE** ($512 \times 512$)**:** When integrated with the DC-AE autoencoder, our model achieves a new SOTA FID score of **1.79** with only 2 NFEs. This result surpasses the leading consistency distillation model, sCD-XXL, which records an FID of 1.88 at 2 NFEs. Notably, our method achieves this superior image quality using a significantly more efficient model with only 675M parameters, compared to the 1.5B parameters of sCD-XXL.

(c) **Performance with VA-VAE** ($256 \times 256$)**:** Using the VA-VAE architecture, our method sets another benchmark, achieving a remarkable FID of **1.48** in just 2 NFEs. This represents a substantial improvement over the best-performing distillation method, IMM, which only reaches an FID of 1.99 after a much more costly $8 \times 2 = 16$ NFEs.

Table 2: **System-level quality comparison for few-step generation task on class-conditional ImageNet-1K.** The **best** results of each resolution are highlighted.

| | 512 × 512 | | | | 256 × 256 | | | | |
|---|---|---|---|---|---|---|---|---|---|
| Method | NFE ↓ | FID ↓ | #Params | #Epochs | Method | NFE ↓ | FID ↓ | #Params | #Epochs |
| **Diffusion & flow-matching Models** | | | | | | | | | |
| ADM-G (Dhariwal & Nichol, 2021) | 250×2 | 7.72 | 559M | 388 | ADM-G (Dhariwal & Nichol, 2021) | 250×2 | 4.59 | 559M | 396 |
| U-ViT-H/4 (Bao et al., 2023) | 50×2 | 4.05 | 501M | 400 | U-ViT-H/2 (Bao et al., 2023) | 50×2 | 2.29 | 501M | 400 |
| DiT-XL/2 (Peebles & Xie, 2023) | 250×2 | 3.04 | 675M | 600 | DiT-XL/2 (Peebles & Xie, 2023) | 250×2 | 2.27 | 675M | 1400 |
| SiT-XL/2 (Ma et al., 2024) | 250×2 | 2.62 | 675M | 600 | SiT-XL/2 (Ma et al., 2024) | 250×2 | 2.06 | 675M | 1400 |
| MaskDiT (Zheng et al., 2023) | 79×2 | 2.50 | 736M | - | MDT (Gao et al., 2023) | 250×2 | 1.79 | 675M | 1300 |
| EDM2-S (Karras et al., 2024) | 63 | 2.56 | 280M | 1678 | REPA-XL/2 (Yu et al., 2024) | 250×2 | 1.96 | 675M | 200 |
| EDM2-L (Karras et al., 2024) | 63 | 2.06 | 778M | 1476 | REPA-XL/2 (Yu et al., 2024) | 250×2 | 1.42 | 675M | 800 |
| EDM2-XXL (Karras et al., 2024) | 63 | 1.91 | 1.5B | 734 | Light.DiT (Yao et al., 2025) | 250×2 | 2.11 | 675M | 64 |
| DiT-XL⊕DC-AE | 250×2 | 2.41 | 675M | 400 | Light.DiT (Yao et al., 2025) | 250×2 | 1.35 | 675M | 800 |
| **GANs** | | | | | | | | | |
| BigGAN (Brock et al., 2018) | 1 | 8.43 | 160M | - | BigGAN (Brock et al., 2018) | 1 | 6.95 | 112M | - |
| StyleGAN (Sauer et al., 2022) | 1×2 | **2.41** | 168M | - | GigaGAN (Kang et al., 2023) | 1 | **3.45** | 569M | - |
| **Masked & autoregressive models** | | | | | | | | | |
| MaskGIT (Chang et al., 2022) | 12 | 7.32 | 227M | 300 | MaskGIT (Chang et al., 2022) | 8 | 6.18 | 227M | 300 |
| VAR-$d36$-s (Tian et al., 2024) | 10×2 | **2.63** | 2.3B | 350 | VAR-$d30$-re (Tian et al., 2024) | 10×2 | **1.73** | 2.0B | 350 |
| **1st-order consistency training & distillation** | | | | | | | | | |
| sCT-M (Lu & Song, 2024) | 1 | 5.84 | 498M | 1837 | Shortcut-XL/2 (Frans et al., 2024) | 1 | 10.6 | 676M | 250 |
| | 2 | 5.53 | 498M | 1837 | | 4 | 7.80 | 676M | 250 |
| sCT-L (Lu & Song, 2024) | 1 | 5.15 | 778M | 1274 | IMM-XL/2 (Zhou et al., 2025) | 1×2 | 7.77 | 675M | 3840 |
| | 2 | 4.65 | 778M | 1274 | | 2×2 | 5.33 | 675M | 3840 |
| sCT-XXL (Lu & Song, 2024) | 1 | 4.29 | 1.5B | 762 | | 4×2 | 3.66 | 675M | 3840 |
| | 2 | 3.76 | 1.5B | 762 | | 8×2 | 2.77 | 675M | 3840 |
| sCD-M (Lu & Song, 2024) | 1 | 2.75 | 498M | 1997 | IMM ($\omega = 1.5$) | 1×2 | 8.05 | 675M | 3840 |
| | 2 | 2.26 | 498M | 1997 | | 2×2 | 3.99 | 675M | 3840 |
| sCD-L (Lu & Song, 2024) | 1 | 2.55 | 778M | 1434 | | 4×2 | 2.51 | 675M | 3840 |
| | 2 | 2.04 | 778M | 1434 | | 8×2 | **1.99** | 675M | 3840 |
| sCD-XXL (Lu & Song, 2024) | 1 | 2.28 | 1.5B | 921 | MeanFlow-XL/2 (Geng et al., 2025) | 1 | 3.43 | 676M | 240 |
| | 2 | **1.88** | 1.5B | 921 | | 2 | 2.93 | 676M | 240 |
| UCGM-XL (Sun et al., 2025) | 1 | 2.63 | 675M | 360 | MeanFlow-XL/2 (longer training) | 2 | 2.20 | 676M | 1000 |
| **RCGM (Ours)** | | | | | | | | | |
| ⊕SD-VAE (Rombach et al., 2022) | 1 | 2.61 | 675M | 360 | ⊕SD-VAE (Rombach et al., 2022) | 1 | 2.13 | 675M | 424 |
| ⊕SD-VAE | 2 | 2.25 | 675M | 360 | ⊕SD-VAE | 2 | 1.92 | 675M | 424 |
| ⊕DC-AE (Chen et al., 2024c) | 1 | 2.45 | 675M | 800 | ⊕VA-VAE (Yao et al., 2025) | 1 | 2.25 | 675M | 424 |
| ⊕DC-AE | 2 | **1.79** | 675M | 800 | ⊕VA-VAE | 2 | **1.48** | 675M | 424 |

*In summary, across multiple autoencoder architectures, our* RCGM *consistently delivers a superior trade-off between sample quality, sampling speed, and model parameter efficiency. It establishes new SOTA results while substantially reducing the computational overhead required for high-fidelity image generation.*

**Validating RCGM on real-world applications.** To assess its practical efficacy, we evaluate RCGM on two demanding real-world tasks: text-to-image generation (App. D.1) and the training of few-step unified multimodal models (App. D.2). Our results demonstrate that RCGM exhibits remarkable performance and versatility across these diverse settings, substantially outperforming existing methods in the computationally constrained, few-step sampling regime.

For instance, in text-to-image synthesis, RCGM attains a GenEval score of $0.85$ with only NFE$= 2$. This marks a significant advance over the previous SOTA, SANA-Sprint (Chen et al., 2025c), which achieves a score of $0.77$, thereby establishing a new benchmark for highly efficient generation.

## 5 CONCLUSION AND LIMITATIONS

In this paper, we introduced RCGM, a unified framework for continuous generative modeling that bridges the gap between multi-step and few-step synthesis. Our key innovation is a novel $N$-th order flow matching objective that improves training stability and significantly boosts performance, especially in few-step regimes. Through extensive experiments on ImageNet-1K, we demonstrated that RCGM establishes a new state of the art across a spectrum of few-step generation settings.

Despite its strong performance, RCGM shares a limitation with contemporary generative models: achieving high-fidelity synthesis in extreme few-step regimes (e.g., 1-NFE) remains an open challenge, particularly for high-resolution imagery. We conjecture that this is partly attributable to the absence of an adversarial objective, which has proven effective for enhancing perceptual quality in other generative paradigms. Consequently, a promising direction for future research is the integration of adversarial training into the RCGM framework to further push the boundaries of sample quality in this challenging setting. We leave this promising avenue for future work.

## ACKNOWLEDGEMENT

This work was supported in part by the National Science and Technology Major Project (No. 2022ZD0115101), NSFC under No. 62576285, the Research Center for Industries of the Future (RCIF) at Westlake University, and the Westlake Education Foundation.

We thank the anonymous reviewers for their insightful comments and suggestions, which significantly improved this paper. We also express our gratitude to Xinyi Shang, Yansen Han, and Zhenglin Cheng for their valuable discussions and feedback.

## ETHICS STATEMENT

This research adheres to the *ICLR Code of Ethics* and is committed to the principles of responsible and transparent scientific inquiry. The study involves no human participants, personal or sensitive data, or any activities requiring approval from an institutional ethics review board. All datasets used are publicly accessible under appropriate licenses, with proper attribution given to their original sources. To promote openness and reproducibility, we provide our implementation code and experimental settings for verification and further development by the research community. We also declare that no conflicts of interest or external funding have influenced the design, execution, or presentation of this work.

## REPRODUCIBILITY STATEMENT

Comprehensive details regarding the datasets, model architectures, optimization settings, and training procedures are provided in Sec. 4.1 of the main paper and in App. D. These materials are designed to facilitate the reliable and transparent reproduction of our results. Additionally, our source code will be made publicly available upon acceptance of the paper.

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

CONTENTS

## A  UTILIZATION OF LARGE LANGUAGE MODELS (LLMS)

In this study, Large Language Models (LLMs) are employed at the sentence level to assist in linguistic refinement. Their use was strictly confined to improving grammatical accuracy and overall readability of the manuscript. All research concepts, methodological designs, experimental processes, and analytical findings remain entirely original and have been solely contributed by the authors.

## B  IMPLEMENTATION DETAILS

---

**Algorithm 1** $N$-th Order RCGM Training Step

---

**Require:** Data distribution $p(\mathbf{x})$, order $N \geq 0$, initial time step $\Delta t$ (we set $\Delta t = 0.01$ in this paper), EMA decay rate $\kappa$.
**Require:** Initial parameters $\boldsymbol{\theta}$, stop gradient operator sg, EMA parameters $\boldsymbol{\theta}^-$.
 1: **Preparation Phase:**
 2: Sample data $\mathbf{x} \sim p(\mathbf{x})$, noise $\mathbf{z} \sim \mathcal{N}(\mathbf{0}, \mathbf{I})$.
 3: Sample current time $t \sim \mathcal{U}(0,1)$ and target time $t_{N+1} \sim \mathcal{U}(0,t)$.
 4: Construct perturbed state $\mathbf{x}_t = t\mathbf{z} + (1-t)\mathbf{x}$.
 5: Compute target velocity $\mathbf{v} = \mathbf{z} - \mathbf{x}$.
 6: Define time trajectory: set $t_0 = t$, $t_1 = t_0 - \Delta t$, and let $\{t_i\}_{i=1}^{N+1}$ be linearly spaced between $t_1$ and $t_{N+1}$.
 7: **Recursive Velocity Estimation:**
 8: Initialize accumulated displacement $\boldsymbol{\Delta} = \mathbf{0}$.
 9: Estimate state at $t_1$: $\mathbf{x}_{t_1} = \mathbf{x}_t - \mathbf{v}\Delta t$.
10: **for** $i = 1$ **to** $N$ **do**
11:     Estimate segment displacement: $\mathbf{d}_i = \boldsymbol{F}_{\boldsymbol{\theta}^-}(\mathbf{x}_{t_i}, t_i, t_{i+1}) \cdot (t_{i+1} - t_i)$.
12:     Update state: $\mathbf{x}_{t_{i+1}} \leftarrow \mathbf{x}_{t_i} + \mathbf{d}_i$.
13:     Accumulate: $\boldsymbol{\Delta} \leftarrow \boldsymbol{\Delta} + \mathbf{d}_i$ {*Note:* $\boldsymbol{\Delta}$ *approximates* $\sum_{i=1}^{N} \int_{t_i}^{t_{i+1}} \mathbf{v}(\mathbf{x}_\tau, \tau)d\tau$}
14: **end for**
15: **Loss Calculation:**
16: Compute model output: $\mathbf{u} = \boldsymbol{F}_{\boldsymbol{\theta}}(\mathbf{x}_t, t, t_{N+1})$.
17: Update $\boldsymbol{\theta}$ by minimizing objective:

$$\mathcal{L}(\boldsymbol{\theta}) = \left\| \mathbf{u} - \text{sg}(\mathbf{u}) + \frac{1}{\Delta t} \left[ \text{sg}(\mathbf{u}) \cdot (t - t_{N+1}) + \boldsymbol{\Delta} \right] - \mathbf{v} \right\|_2^2$$

18: Update EMA parameters: $\boldsymbol{\theta}^- \leftarrow \kappa\boldsymbol{\theta}^- + (1-\kappa)\boldsymbol{\theta}$.

---

During the inference stage, our sampling strategy is designed for simplicity and independence from hyperparameter tuning. For one-step generation, we follow the direct prediction procedure described in Sec. 2, requiring no external hyperparameter configuration. For few-step and multi-step generation, we adopt the UCGM sampler (Sun et al., 2025) using its default settings without any additional manual tuning.

### B.1  QUALITATIVE ANALYSIS OF RCGM TRAINING DYNAMICS

We investigate the convergence properties of the proposed $N$-th Order RCGM by modeling the training process as a dynamic interplay between a *Ground-Truth Anchor* and a *Recursive Bootstrap Tail*. The student model aims to match a hybrid target displacement $\mathbf{D}_{\text{target}}$ constructed over the interval $[t, t_{N+1}]$. Based on Alg. 1, this target can be conceptually decomposed as:

$$\mathbf{D}_{\text{target}} \approx \underbrace{\mathbf{v} \cdot \Delta t}_{\substack{\text{Anchor} \\ \text{(Low Bias, High Variance)}}} + \underbrace{\sum_{i=1}^{N} \boldsymbol{F}_{\boldsymbol{\theta}^-}(\mathbf{x}_{t_i}, t_i, t_{i+1}) \cdot (t_i - t_{i+1})}_{\substack{\text{Recursive Tail} \\ \text{(High Bias, Low Variance)}}} \tag{10}$$

Here, the **Anchor** is derived from the real data distribution (providing unbiased instantaneous direction), while the **Tail** is generated by the EMA teacher $\boldsymbol{\theta}^-$ (providing smooth but potentially biased manifold approximations). The stability of this system depends critically on the interaction between the recursive order $N$ and the EMA decay rate $\kappa$. We identify two distinct operating regimes:

**Regime I: The Error Accumulation Regime ($N \geq 2$).** When the recursive order $N$ is high, the target trajectory is dominated by the model's own predictions (the Tail). In this regime, the primary challenge is **recursive variance amplification**. Since the estimation at step $i$ depends on the state $\mathbf{x}_{t_i}$ predicted at step $i - 1$, small perturbations in the teacher's parameters $\boldsymbol{\theta}^-$ are compounded exponentially through the recursive chain. This phenomenon is analogous to error propagation in numerical integration. If the teacher evolves too quickly (i.e., low $\kappa$), the target values become non-stationary and noisy, preventing the student from converging. Consequently, high-order training requires strong stabilization: we must employ a high EMA decay rate (e.g., $\kappa \approx 0.999$) to effectively "freeze" the teacher, minimizing the temporal variance of the target and ensuring that the recursive tail provides a consistent guidance signal.

**Regime II: The Bias Correction Regime ($N = 1$).** When $N = 1$, the recursive tail is short, and the dominant error source shifts from variance to **geometric bias**. A single-step linear approximation inevitably undershoots the curvature of the true data manifold. If the teacher is too stable (i.e., $\kappa \approx 1$), the model risks converging to a spurious fixed point where the student simply mimics the teacher's biased linear prediction, ignoring the curvature information. Crucially, the **Anchor** term $\mathbf{v} \cdot \Delta t$ contains the necessary first-order derivative information to correct this bias. To effectively incorporate this correction, the system requires "plasticity": the teacher must update rapidly to reflect the rectified trajectory suggested by the Anchor. Thus, low-order training necessitates a lower EMA decay rate (e.g., $\kappa \approx 0.90$). This increases the system's responsiveness, allowing the Anchor to actively correct the Tail's geometric errors and preventing the solidification of incorrect linear assumptions.

**Synthesis.** Our analysis reveals a fundamental trade-off between **stability** and **plasticity**. High $N$ enables better long-range approximation but introduces instability, necessitating a rigid teacher (High $\kappa$). Low $N$ is stable but geometrically biased, necessitating an adaptive teacher (Low $\kappa$) to drive correction. This informs our hyperparameter selection strategy: $\kappa$ should be positively correlated with the recursive order $N$.

## C  RELATED WORK

The landscape of continuous-time generative models has evolved from multi-step integration towards high-fidelity, few-step synthesis. Our work builds upon this trajectory by addressing the limitations of existing paradigms. We contextualize our contributions by surveying the two dominant research thrusts that enable rapid generation: interval-based consistency and adversarial refinement.

### C.1  FOUNDATIONS: MULTI-STEP INTEGRATION OF INSTANTANEOUS FIELDS

The dominant paradigm in continuous generative modeling, including Denoising Diffusion Models (Ho et al., 2020; Song et al., 2020b) and Flow-Matching (Lipman et al., 2022), is the learning of an *instantaneous* velocity field. These models train a neural network to approximate the local dynamics $\frac{\mathrm{d}\mathbf{x}_t}{\mathrm{d}t}$ of a Probability Flow Ordinary Differential Equation (PF-ODE). To generate a sample, one must numerically integrate this field, typically requiring hundreds or thousands of steps to ensure fidelity. The core limitation of this approach is its sensitivity to coarse discretization; when using few steps, large truncation errors accumulate, particularly for trajectories with high curvature, leading to a significant degradation in sample quality (Karras et al., 2022). This challenge has catalyzed the development of methods designed for the few-step regime.

### C.2  INTERVAL-BASED CONSISTENCY FOR FEW-STEP GENERATION

A major research thrust aims to overcome this limitation by enforcing consistency over finite time intervals, effectively teaching the model about the integrated structure of the ODE path. Consistency Models (CMs) (Song et al., 2023) pioneered this approach by enforcing a *relative* constraint: the

model's prediction of the trajectory's endpoint ($\mathbf{x}_0$) should be consistent across different starting points ($\mathbf{x}_t, \mathbf{x}_{t-\Delta t}$) on the same path. This concept was extended by methods like MeanFlow (Geng et al., 2025), which directly model their proposed average velocity to predict other points beyond the endpoint along the PF-ODE.

However, a critical implementation challenge emerged: the need to compute time derivatives to enforce these interval-based objectives. Early methods relied on Jacobian-Vector Products (JVP) (Geng et al., 2025; Lu & Song, 2024), which introduced a severe scalability bottleneck. JVP is computationally intensive and, more importantly, incompatible with essential modern training optimizations like FlashAttention (Dao et al., 2022) and FSDP-based distributed training (Zhao et al., 2023), hindering its application to billion-parameter models. Consequently, the field has pivoted to using finite-difference estimators as a scalable and hardware-friendly alternative (Sun et al., 2025). These estimators, which rely only on forward passes, ensure compatibility with contemporary large-scale training infrastructures.

### C.3 Adversarial Refinement for One-Step Synthesis

A parallel and complementary approach achieves high-fidelity, one-step generation by incorporating external, adversarial signals. This is motivated by the fact that relative consistency constraints do not explicitly guarantee that the final output lies on the true data manifold. Adversarial objectives provide an *absolute* anchor to the data distribution.

Methods in this family, such as distillation techniques like DMD/DMD2 (Yin et al., 2024b;a) and other GAN-based refiners (Sauer et al., 2024b;a; Zheng et al., 2025), employ an auxiliary discriminator to sharpen model outputs. This adversarial pressure can be powerful enough to enable a distilled "student" model to surpass the performance of its "teacher." However, this reliance is a double-edged sword. It often introduces training instability and increases computational overhead due to the auxiliary network. Critically, these frameworks typically depend on a frozen, pre-trained teacher to generate a large dataset of target samples. For ultra-large models, the cost of generating this dataset can be prohibitive, in some cases exceeding the cost of training the student model itself (Yin et al., 2024a). This trade-off between sample fidelity and training complexity remains a key challenge.

## D Detailed Experiment

### D.1 Comparison with Text-to-image Models

To validate the real-world applicability of our approach, we benchmarked RCGM on the text-to-image synthesis task, presenting detailed results in Tab. 3. For this evaluation, we fine-tuned the SANA-0.6B and SANA-1.6B backbones for $30{,}000$ steps, using batch sizes of $128$ and $64$, respectively. The experimental results clearly demonstrate that RCGM achieves SOTA performance while operating with an extremely low NFE. We conduct all experiments on publicly available datasets (Chen et al., 2025d; Ye et al., 2025) and models (Chen et al., 2025c; Xie et al., 2024a) to ensure reproducibility and transparency.

(a) **SOTA quality at 2-NFE:** With the addition of a second inference step, RCGM's generative quality is further enhanced, reaching a GenEval score of 0.85 for the 0.6B model and 0.84 for the 1.6B version. This level of performance surpasses not only the leading few-step models but also powerful multi-step architectures such as SANA-1.5 (0.81) and Playground v3 (0.76). This top-tier output is delivered with a highly competitive throughput of 6.50 samples/s and a latency of just 0.26s.

(b) **Superiority in the 1-NFE setting:** When constrained to a single inference step—a challenging setting for generative models—RCGM markedly outperforms its peers. Our 0.6B variant achieves a GenEval score of 0.80, placing it ahead of strong contenders like SANA-Sprint-1.6B (0.76) and FLUX-Schnell (0.69). Crucially, this high-quality output is paired with unmatched efficiency; at 7.30 samples/s, RCGM-0.6B stands as the fastest model in this category.

*The success of* RCGM *is especially compelling given its fundamental simplicity. Powerful baselines like SANA-Sprint employ a sophisticated hybrid loss, integrating sCM (Lu & Song, 2024) with LADD (Sauer et al., 2024a)—an adversarial technique requiring a dedicated discriminator. Our approach, however, relies solely on the straightforward objective in (8). The fact that this minimalist*

Table 3: **System-level benchmark of our RCGM against text-to-image models.** Throughput (batch=10) and latency (batch=1) were measured on a single A100 (BF16). The **best** and second-best results among few-step models are highlighted. [†]Our evaluation.

| Method | NFE ↓ | Throughput ↑ (samples/s) | Latency (s) ↓ | #Params | GenEval ↑ | DPG-Bench ↑ |
|---|---|---|---|---|---|---|
| *Multi-step models* | | | | | | |
| SDXL (Podell et al., 2023) | 50×2 | 0.15 | 6.5 | 2.6B | 0.55 | 74.7 |
| PixArt-Σ (Chen et al., 2024a) | 20×2 | 0.40 | 2.7 | 0.6B | 0.54 | 80.5 |
| SD3-Medium (Esser et al., 2024b) | 28×2 | 0.28 | 4.4 | 2.0B | 0.62 | 84.1 |
| FLUX-Dev (Labs, 2024) | 50×2 | 0.04 | 23.0 | 12.0B | 0.67 | 84.0 |
| Playground v3 (Liu et al., 2024) | - | 0.06 | 15.0 | 24B | 0.76 | 87.0 |
| SANA-0.6B (Xie et al., 2024a) | 20×2 | 1.7 | 0.9 | 0.6B | 0.64 | 83.6 |
| SANA-1.6B (Xie et al., 2024a) | 20×2 | 1.0 | 1.2 | 1.6B | 0.66 | 84.8 |
| SANA-1.5 (Xie et al., 2025a) | 20×2 | 0.26 | 4.2 | 4.8B | 0.81 | 84.7 |
| Lumina-Image-2.0 (Qin et al., 2025) | 18×2 | - | - | 2.6B | 0.73 | 87.2 |
| *Few-step models* | | | | | | |
| SDXL-DMD2 (Yin et al., 2024a) | 2 | 2.89 | 0.40 | 0.9B | 0.58 | - |
| FLUX-Schnell (Labs, 2024) | 2 | 0.92 | 1.15 | 12.0B | 0.71 | - |
| SANA-Sprint-0.6B (Chen et al., 2025c) | 2 | 6.46 | 0.25 | 0.6B | 0.76 | 81.5[†] |
| SANA-Sprint-1.6B (Chen et al., 2025c) | 2 | 5.68 | 0.24 | 1.6B | 0.77 | 82.1[†] |
| SDXL-LCM (Luo et al., 2023) | 2 | 2.89 | 0.40 | 0.9B | 0.44 | - |
| PixArt-LCM (Chen et al., 2024b) | 2 | 3.52 | 0.31 | 0.6B | 0.42 | - |
| PCM (Wang et al., 2024) | 2 | 2.62 | 0.56 | 0.9B | 0.55 | - |
| SD3.5-Turbo (Esser et al., 2024a) | 2 | 1.61 | 0.68 | 8.0B | 0.53 | - |
| PixArt-DMD (Chen et al., 2024a) | 1 | 4.26 | 0.25 | 0.6B | 0.45 | - |
| SDXL-DMD2 (Yin et al., 2024a) | 1 | 3.36 | 0.32 | 0.9B | 0.59 | - |
| FLUX-Schnell (Labs, 2024) | 1 | 1.58 | 0.68 | 12.0B | 0.69 | - |
| SANA-Sprint-0.6B (Chen et al., 2025c) | 1 | 7.22 | 0.21 | 0.6B | 0.72 | 78.6[†] |
| SANA-Sprint-1.6B (Chen et al., 2025c) | 1 | 6.71 | 0.21 | 1.6B | 0.76 | 80.1[†] |
| SDXL-LCM (Luo et al., 2023) | 1 | 3.36 | 0.32 | 0.9B | 0.28 | - |
| PixArt-LCM (Chen et al., 2024b) | 1 | 4.26 | 0.25 | 0.6B | 0.41 | - |
| PCM (Wang et al., 2024) | 1 | 3.16 | 0.40 | 0.9B | 0.42 | - |
| SD3.5-Turbo (Esser et al., 2024a) | 1 | 2.48 | 0.45 | 8.0B | 0.51 | - |
| TiM (Wang et al., 2025b) | 1 | - | - | 0.8B | 0.67 | 75.0 |
| **RCGM-0.6B (Ours)** | 2 | 6.50 | 0.26 | 0.6B | **0.85** | 80.3 |
| **RCGM-1.6B (Ours)** | 2 | 5.71 | 0.25 | 1.6B | **0.84** | 79.1 |
| **RCGM-0.6B (Ours)** | 1 | 7.30 | 0.23 | 0.6B | **0.80** | 77.2 |
| **RCGM-1.6B (Ours)** | 1 | 6.75 | 0.22 | 1.6B | **0.78** | 76.5 |

*framework yields SOTA results demonstrates that* RCGM *offers a more elegant and direct solution to the enduring conflict between sampling speed and visual fidelity.*

## D.2 COMPARISON WITH UNIFIED MULTIMODAL MODELS

The development of Unified Multimodal Models (UMM), which are capable of both profound comprehension (typically yielding textual outputs) and sophisticated generation (resulting in visual outputs), represents a significant frontier in artificial intelligence. Such integrated systems hold the potential to unlock synergistic capabilities, where understanding informs generation and vice versa, leading to more intelligent and versatile applications (Google, 2025a;b; OpenAI, 2025).

Recent advancements in UMMs have showcased their considerable potential across a diverse range of applications, including high-fidelity text-to-image generation and intricate image editing (Wu et al., 2025a; Pan et al., 2025). These models have been lauded within the research community for their powerful generative abilities (Chen et al., 2025a; Dong et al., 2024).

However, a primary obstacle to the widespread adoption of these models is their prohibitive computational cost. This inefficiency stems from their reliance on iterative, diffusion-based generation processes, which incur significant overhead and lead to slow inference times. To address this critical efficiency bottleneck, we integrate our proposed method, RCGM, into SOTA UMMs.

We demonstrate this by fine-tuning two prominent open-source models: first, conducting full-parameter tuning on OpenUni-L-512 (Wu et al., 2025c) for $60,000$ steps with a batch size of $128$; and second, applying parameter-efficient LoRA (Hu et al., 2022) tuning (with $r = 64$ and $\alpha = 64$) to Qwen-Image-20B Wu et al. (2025a) for $7,000$ steps with a batch size of $64$. The experimental results

Table 4: **System-level comparison of RCGM with unified multimodal models on generation tasks.** Results compare inference efficiency (NFE) and performance across three benchmarks. Best and second-best scores are highlighted as **bold** and underline, respectively. [†] indicates results using LLM-rewritten prompts on GenEval. All our experiments were conducted on $8\times$ NVIDIA H800 GPUs.

| Method | NFE ↓ | Image Generation | | |
|---|---|---|---|---|
| | | GenEval ↑ | DPG-Bench ↑ | WISE ↑ |
| Show-o2-7B (Xie et al., 2025b) | $50\times2$ | 0.76 | 86.14 | 0.39 |
| OmniGen (Xiao et al., 2024) | $50\times2$ | 0.70 | 81.16 | - |
| OmniGen2 (Wu et al., 2025b) | $50\times2$ | $0.80 / 0.86^{\dagger}$ | 83.57 | - |
| Show-o (Xie et al., 2024b) | $50\times2$ | 0.68 | 67.27 | 0.35 |
| Janus-Pro (Chen et al., 2025e) | - | 0.80 | 84.19 | 0.35 |
| MetaQuery-XL (Pan et al., 2025) | $30\times2$ | $0.78 / 0.80^{\dagger}$ | 81.10 | 0.55 |
| BLIP3-o-8B (Chen et al., 2025b) | $30\times2 + 50\times2$ | 0.84 | 81.60 | **0.62** |
| UniWorld-V1 (Lin et al., 2025) | $28\times2$ | $0.80 / 0.84^{\dagger}$ | - | 0.55 |
| OpenUni-L-512 (Wu et al., 2025c) | $20\times2$ | 0.85 | 81.54 | 0.52 |
| Bagel (Deng et al., 2025) | $50\times2$ | $0.82 / 0.88^{\dagger}$ | | 0.52 |
| Qwen-Image-20B (Wu et al., 2025a) | $50\times2$ | **0.87** | **88.32** | **0.62** |
| OpenUni-L-512⊕CM (Song et al., 2023) (model collapse) | 2 | 0.0 | - | - |
| OpenUni-L-512⊕CM (Song et al., 2023) (model collapse) | 1 | 0.0 | - | - |
| Qwen-Image-20B⊕CM (Song et al., 2023) (model collapse) | 2 | 0.0 | - | - |
| Qwen-Image-20B⊕CM (Song et al., 2023) (model collapse) | 1 | 0.0 | - | - |
| Qwen-Image-20B⊕MeanFlow (Geng et al., 2025) (out of memory) | 2 | - | - | - |
| Qwen-Image-20B⊕MeanFlow (Geng et al., 2025) (out of memory) | 1 | - | - | - |
| OpenUni-L-512⊕RCGM (ours) | 2 | 0.85 | 80.15 | 0.50 |
| OpenUni-L-512⊕RCGM (ours) | 1 | 0.80 | 76.40 | 0.45 |
| Qwen-Image-20B⊕RCGM (ours) | 8 | **0.87** | 87.39 | 0.58 |
| Qwen-Image-20B⊕RCGM (ours) | 2 | 0.82 | 84.09 | 0.50 |
| Qwen-Image-20B⊕RCGM (ours) | 1 | 0.52 | 59.50 | 0.30 |

presented in Tab. 4 clearly demonstrate our method's efficacy. Specifically, we observe the following key outcomes:

(a) **Significant reduction in computational cost:** Our method dramatically reduces the NFE to just 1 or 2, a stark contrast to the 40 to 100 NFEs required by the original models. This represents a reduction of over 95% in computational workload, thereby enabling faster and more efficient image generation.

(b) **Competitive performance with fewer steps:** When applied to *OpenUni-L-512*, our method with an NFE of 2 achieves a GenEval score of 0.85, matching the performance of the original model which requires 40 steps. While there is a slight decrease in the DPG-Bench and WISE scores, the performance remains highly competitive. Even with a single step (NFE=1), our model maintains a strong GenEval score of 0.80.

(c) **Effective application to larger models:** With the more powerful *Qwen-Image-20B*, our method at 2-NFE achieves a GenEval score of 0.82 and a DPG-Bench score of 84.09. Although these scores are slightly lower than the original model's 100-NFE process, they are still comparable to other leading UMMs that require significantly more computational resources. This demonstrates the scalability and effectiveness of our approach on larger, more capable models.

In summary, our proposed method provides a compelling solution to the efficiency challenges inherent in diffusion-based UMMs. By substantially decreasing the required number of generation steps while preserving a high level of performance, RCGM paves the way for more practical and accessible applications of these powerful multimodal systems.

**Discussion on open-source community efforts.** To the best of our knowledge, Qwen-Image-Lightning (ModelTC, 2025) represents the sole open-source initiative focused on training a few-step variant of a large-scale UMM. This method is based on the Distribution Matching Distillation (DMD2) framework (Yin et al., 2024a); however, it notably omits its generative adversarial network (GAN) loss component. This crucial omission, however, directly leads to a significant and widely acknowledged problem: **generation pattern collapse**. Specifically, Qwen-Image-Lightning is known to suffer from generating highly similar, or even nearly identical, images across diverse input prompts, severely limiting its generative diversity and overall practical utility.

Table 5: **Quantitative comparison of RCGM against baselines on Qwen-Image-20B for text-to-image generation.** *raw* denotes full-parameter implementations for the generator and scoring networks, which incur excessive GPU memory costs. Consequently, to enable baseline methods (VSD, SiD, and DMD) to run within memory constraints, we implement their fake score networks using LoRA ($r$=64).

| Method | NFE $\downarrow$ | Image Generation | | |
|---|---|---|---|---|
| | | GenEval $\uparrow$ | DPG-Bench $\uparrow$ | WISE $\uparrow$ |
| VSD (Wang et al., 2023) (*raw*, out of GPU memory) | N/A | - | - | - |
| DMD (Yin et al., 2024b) (*raw*, out of GPU memory) | N/A | - | - | - |
| DMD2 (Yin et al., 2024a) (*raw*, out of GPU memory) | N/A | - | - | - |
| SiD (Zhou et al., 2024) (*raw*, out of GPU memory) | N/A | - | - | - |
| VSD (Wang et al., 2023) | 1 | 0.67 | 84.44 | 0.22 |
| | 2 | 0.73 | 86.16 | 0.34 |
| DMD (Yin et al., 2024b) | 1 | 0.81 | 84.31 | 0.47 |
| | 2 | 0.80 | 84.08 | 0.46 |
| SiD (Zhou et al., 2024) | 1 | 0.77 | 87.05 | 0.42 |
| | 2 | 0.78 | 86.94 | 0.41 |
| sCM (Lu & Song, 2024) (JVP-free) | 1 | 0.55 | 79.12 | 0.21 |
| | 2 | 0.64 | 83.91 | 0.46 |
| MeanFlow (Geng et al., 2025) (JVP-free) | 1 | 0.49 | 82.39 | 0.33 |
| | 2 | 0.57 | 85.09 | 0.40 |
| RCGM ($N$=1) | 1 | 0.47 | 74.31 | 0.19 |
| | 2 | 0.71 | 81.48 | 0.42 |
| RCGM ($N$=2) | 1 | 0.55 | 75.56 | 0.33 |
| | 2 | 0.78 | 85.27 | 0.51 |

### D.3 QUANTITATIVE ANALYSIS ON QWEN-IMAGE-20B

Tab. 5 presents a comprehensive quantitative comparison of RCGM against SOTA acceleration methods on the Qwen-Image-20B model. The experiments were conducted on 8 GPUs with a local batch size of 4. Unless otherwise specified, models were trained for 3,000 steps with an AdamW learning rate of $1 \times 10^{-5}$. Our analysis highlights three key observations regarding scalability, order capability, and compatibility with adversarial objectives.

**Scalability to Full-Parameter Training on 20B Models.** A significant challenge in accelerating large-scale diffusion models is the excessive memory consumption and training instability associated with Consistency Model (CM) based approaches. As shown in the top section of Tab. 5, baseline methods such as VSD, DMD, and SiD fail to run in the "raw" full-parameter setting due to Out-Of-Memory (OOM) errors, forcing them to rely on low-rank approximations (LoRA, $r = 64$) to fit within memory constraints. In contrast, RCGM demonstrates superior memory efficiency and stability, successfully enabling **full-parameter training** on the 20B parameter model. This result underscores the robustness of our method, overcoming the well-known instability issues often plaguing CM-series methods in high-parameter regimes.

**Impact of Higher-Order Trajectory Approximation.** We observe a clear performance correlation with the approximation order $N$. While the first-order variant ($N = 1$) provides a baseline acceleration, increasing the order to $N = 2$ yields a substantial performance leap. Specifically, RCGM ($N = 2$) significantly outperforms the first-order counterpart across all metrics. For instance, at NFE=2, the second-order model achieves a GenEval score of 0.78 compared to 0.71 for $N = 1$.

## E THEORETICAL ANALYSIS

### E.1 A RECURSIVE LEARNING PERSPECTIVE OF CONSISTENCY MODELS

The consistency model training objective enforces self-consistency along the sampling trajectory. Given the parameterization $\boldsymbol{f}^{\mathbf{x}}(\boldsymbol{F}_t, \mathbf{x}_t, t) := \frac{\alpha(t) \cdot \boldsymbol{F}_t - \hat{\alpha}(t) \cdot \mathbf{x}_t}{\alpha(t) \cdot \hat{\gamma}(t) - \hat{\alpha}(t) \cdot \gamma(t)}$, the objective is formulated as:

$$\mathbb{E}_{\mathbf{x}_t, t} \left[ d \left( \boldsymbol{f}^{\mathbf{x}}(\boldsymbol{F}_t, \mathbf{x}_t, t), \text{stopgrad}(\boldsymbol{f}^{\mathbf{x}}(\boldsymbol{F}_{t-\Delta t}, \mathbf{x}_{t-\Delta t}, t - \Delta t)) \right) \right].$$

We focus on the specific case of flow matching (Lipman et al., 2022), where $\alpha(t) = t$, $\gamma(t) = 1 - t$, $\hat{\alpha}(t) = 1$, and $\hat{\gamma}(t) = -1$.

Under these conditions, the training loss $\mathcal{L}_{\mathrm{CM}}(\boldsymbol{\theta})$ simplifies to:

$$\mathcal{L}_{\mathrm{CM}}(\boldsymbol{\theta}) = d\left(t \cdot \boldsymbol{F}_{\boldsymbol{\theta}^-}(\mathbf{x}_t) - \mathbf{x}_t, (t - \Delta t) \cdot \boldsymbol{F}_{\boldsymbol{\theta}^-}(\mathbf{x}_{t-\Delta t}) - \mathbf{x}_{t-\Delta t}\right). \tag{11}$$

This objective minimizes the distance between the current model prediction and the target prediction derived from the preceding time step.

We can express the $\ell_2$ loss explicitly:

$$\mathcal{L}_{\mathrm{CM}}(\boldsymbol{\theta}) = \|t \cdot \boldsymbol{F}_{\boldsymbol{\theta}^-}(\mathbf{x}_t) - \mathbf{x}_t - (t - \Delta t) \cdot \boldsymbol{F}_{\boldsymbol{\theta}^-}(\mathbf{x}_{t-\Delta t}) + \mathbf{x}_{t-\Delta t}\|_2^2. \tag{12}$$

To analyze the training dynamics, we consider the limit $\Delta t \to 0$ and apply a Taylor expansion, which yields:

$$\mathcal{L}_{\mathrm{CM}}(\boldsymbol{\theta}) = \left\|t \cdot \boldsymbol{F}_{\boldsymbol{\theta}^-}(\mathbf{x}_t) - (t - \Delta t) \cdot \boldsymbol{F}_{\boldsymbol{\theta}^-}(\mathbf{x}_{t-\Delta t}) - \frac{d\mathbf{x}_t}{dt} \cdot \Delta t\right\|_2^2. \tag{13}$$

Minimizing this loss corresponds to the following update rule:

$$t \cdot \boldsymbol{F}_{\boldsymbol{\theta}^-}(\mathbf{x}_t) \leftarrow (t - \Delta t) \cdot \boldsymbol{F}_{\boldsymbol{\theta}^-}(\mathbf{x}_{t-\Delta t}) + \frac{d\mathbf{x}_t}{dt} \cdot \Delta t. \tag{14}$$

By induction, the model learns the integrated velocity field:

$$t \cdot \boldsymbol{F}_{\boldsymbol{\theta}^-}(\mathbf{x}_t) \leftarrow \int_0^t \frac{d\mathbf{x}_\tau}{d\tau} d\tau = \mathbf{x}_t - \mathbf{x}_0. \tag{15}$$

Let us define the prediction function as $\boldsymbol{f}(\mathbf{x}_t, 0) := \mathbf{x}_t - \mathbf{x}_0$. Substituting this into (14), we obtain the following recursive relationship:

$$\boxed{\boldsymbol{f}(\mathbf{x}_t, 0) \leftarrow \boldsymbol{f}(\mathbf{x}_{t-\Delta t}, 0) + \frac{d\mathbf{x}_t}{dt} \cdot \Delta t}. \tag{16}$$

This result confirms that the consistency model training objective is equivalent to recursively learning the velocity field of the underlying ODE.

