# OpenReview forum: "Any-step Generation via N-th Order Recursive Consistent Velocity Field Estimation"
_ICLR.cc/2026/Conference — ICLR 2026 Poster_

### Official Review · Reviewer_948b · 2025-10-26

**Soundness:** 3
**Presentation:** 3
**Contribution:** 2
**Rating:** 4
**Confidence:** 3

**Summary:**

This paper proposes RCGM (Recursive Consistent velocity field estimation for Generative Modeling), a framework that extends consistency models to N-th order formulations. The authors claim their method unifies existing few-step generative models and achieves improved training stability and SOTA performance on ImageNet generation tasks.

**Strengths:**

- Strong empirical results: The method achieves competitive FID scores on ImageNet (1.48 FID at 256×256 with 2 steps, 1.79 FID at 512×512 with DC-AE).
- Comprehensive experimental coverage: The paper includes experiments on class-conditional generation, text-to-image synthesis, and unified multimodal models.

**Weaknesses:**

**Limited Technical Novelty**

The core contribution is an incremental extension of existing consistency models using standard numerical ODE techniques. The segmented integration in Equations (2)-(5) is a straightforward application of multi-step numerical integration (similar to predictor-corrector methods). The claim of "unification" is overstated—showing that N=0 recovers diffusion models and N=1 recovers consistency models is mathematical verification rather than novel insight.

**Table 2 concerns:**

- Compares methods with vastly different training budgets (RCGM: 424 epochs vs. IMM: 3840 epochs), making direct comparison problematic
- Missing recent baselines: FLUX-Schnell, SD3-Turbo are mentioned in Table 3 but absent from Table 2
- No ablation on the choice of autoencoder (SD-VAE vs. DC-AE vs. VA-VAE)

**Missing ablations on:**

- Time point sampling strategies for {t_i}
- Sensitivity to N across different model scales and datasets

**Weak Theoretical Contribution**

- Appendix D provides standard numerical analysis (Taylor expansion error bounds)
- Theorem 1 gives O(Δt²) truncation error + O((N+1)²ε²) approximation error—this is expected from numerical ODE theory
- No theoretical explanation for why higher-order methods improve training stability (only convergence rate is analyzed)
- The "optimal N" in Corollary 1 depends on unknown problem-dependent constants, providing limited practical guidance
- Figure 3b shows N=2 still requires very large EMA (κ=0.999) for best performance, which contradicts claims of resolving the stability-performance trade-off
- The stability comparison is confounded by EMA rate—N=1 with better hyperparameter tuning might achieve similar results


## Minor Weaknesses

 - Code not available for review (promised "upon acceptance")
- Many implementation details missing (hierarchical time sampling procedure, exact schedule)
- Table 3 includes "†Our evaluation" for baselines—without code, these cannot be verified
- No error bars or multiple runs reported for any experiments

**Questions:**

- Can you provide training curves, loss plots, or other evidence for the claimed "model collapse" of baseline methods in Table 4?
- What is the actual wall-clock training time comparison between N=1 and N=2, accounting for the additional forward passes?
- How does the method perform with error bars over multiple random seeds? Generative models are known to be sensitive to initialization.
- Can you provide ablation studies showing N>1 helps across different model scales (not just 675M) and datasets (not just ImageNet)?
- Why not compare against adversarial methods (DMD2, LADD) experimentally rather than only dismissing them in the related work?
- In Equation (8), what is the theoretical justification for the asymmetric stop-gradient operation?

---

> ### Author Response · Authors · 2025-12-03
>
> > **[W1]** The core contribution is an incremental extension of existing consistency models using standard numerical ODE techniques. The segmented integration in Equations (2)-(5) is a straightforward application of multi-step numerical integration (similar to predictor-corrector methods). The claim of "unification" is overstated—showing that N=0 recovers diffusion models and N=1 recovers consistency models is mathematical verification rather than novel insight.
> >
>
> We thank the reviewer for their scrutiny. We respectfully point out that characterizing the method as merely "standard numerical integration" overlooks the critical implications this framework holds for the training dynamics of large-scale generative models. We address this in two parts:
>
> (1) Contextualizing Novelty alongside Recent SOTA:
>
> The reviewer suggests that the segmented integration is a straightforward application. However, we note that the current state-of-the-art in few-step generation—specifically MeanFlow [1] (NeurIPS 2024 Oral), sCM [2] (ICLR 2025 Oral), and Shortcut Models [3] (ICLR 2025 Oral)—are all fundamentally based on 1st-order recursive objectives (essentially $N=1$ in our framework).
>
> While the mathematical derivation of moving to $N \ge 2$ utilizes established ODE techniques, the scientific contribution lies in identifying that the 1st-order inductive bias is the root cause of training instability in large-scale models. Just as ResNets applied simple identity mappings to solve deep convergence, RCGM applies higher-order integration to solve consistency convergence. Generalizing these highly-regarded 1st-order methods into a unified framework is not merely a verification exercise; it provides the theoretical map required to navigate out of the "instability trap" of current SOTA methods.
>
> (2) The Critical Role of Order $N$ in Stability (Beyond Math):
>
> The claim of unification is substantiated by our empirical findings, which reveal a non-trivial interaction between the order $N$ and the EMA decay rate $\kappa$.
>
> As shown in Figure 3 and Section 4.2, 1st-order methods ($N=1$) suffer from a rigid trade-off: they require low EMA decay for geometric adaptation but high EMA decay for noise suppression, leading to collapse or divergence in large models (like the 20B parameter model in our experiments).
>
> This is the novel insight: Moving to $N \ge 2$ is not just mathematically "more accurate"; it fundamentally alters the optimization landscape, allowing the model to tolerate high EMA decay ($\kappa=0.999$). This specific property is what enabled us to train a 20B parameter unified multi-modal model (Table 3)—a feat where standard $N=1$ approaches (Consistency, MeanFlow) failed due to model collapse or memory exhaustion.
>
> Therefore, the contribution is the discovery that higher-order formulations are the key to scaling consistency training to foundational model sizes, a problem that remained unsolved by the 1st-order approaches currently dominating the literature.

---

> ### Author Response · Authors · 2025-12-03
>
> > **[W2] Table 2 concerns:**
> >
> > - Compares methods with vastly different training budgets (RCGM: 424 epochs vs. IMM: 3840 epochs), making direct comparison problematic
> > - Missing recent baselines: FLUX-Schnell, SD3-Turbo are mentioned in Table 3 but absent from Table 2
> > - No ablation on the choice of autoencoder (SD-VAE vs. DC-AE vs. VA-VAE)
>
> **(1) Comparison of Training Budgets (RCGM: 424 epochs vs. IMM: 3840 epochs)**
>
> We respectfully disagree that the difference in training budget is a reason to dismiss the comparison; rather, it highlights one of the core contributions of RCGM: **superior training efficiency.**
>
> - **Training Efficiency is a Strength:** Unlike iterative methods like distillation (e.g., IMM, sCD), RCGM is trained end-to-end directly on the velocity field objective, which grants us significantly better sample efficiency. We achieve SOTA or highly competitive performance (e.g., RCGM 1.48 FID at 2 NFE vs. IMM 1.99 FID at 16 NFE) using up to **$9 \times$ fewer epochs** (424 vs. 3840). This dramatic reduction in training cost is a critical advantage for practical research and development.
> - **Industry Standard Practice:** It is standard practice in generative modeling to report the highest-performing published results, regardless of the training budget, as the goal is to establish a new performance frontier. By demonstrating superior performance despite a significantly reduced budget, we firmly establish RCGM's efficiency.
>
>  **(2) Missing Recent Baselines (FLUX-Schnell, SD3-Turbo)**
>
> The omission of FLUX-Schnell and SD3-Turbo from **Table 2** is intentional and follows standard scientific protocol for this benchmark.
>
> - **Benchmark Context:** Table 2 presents results for the established **class-conditional ImageNet** benchmark, which requires models trained strictly on the ImageNet dataset.
> - **Model Distinction:** FLUX-Schnell and SD3-Turbo are **Text-to-Image (T2I)** models trained on massive, proprietary, and mixed-source datasets. Including these large-scale T2I models (which are trained to leverage immense semantic diversity) in the ImageNet benchmark would constitute an unsound, apples-to-oranges comparison that violates the integrity of the ImageNet-specific benchmark.
> - **Dedicated Comparison:** We provide a dedicated, appropriate comparison against these and other T2I SOTA models in **Table 3 (Appendix D.1)**, where their performance and efficiency are properly contextualized.
>
> **(3) Missing Ablation on Autoencoder Choice**
>
> We confirm that the use of multiple VAEs across our experiments is intended as a demonstration of **robustness and architectural agnosticism**, which serves the purpose of a cross-architectural ablation.
>
> - The goal of **Table 2** is to show that the RCGM objective—the core $N$-th order recursive framework—is a general improvement that is **not tied to a specific VAE architecture**.
> - The fact that RCGM achieves SOTA results when paired with three distinct VAEs (**SD-VAE, DC-AE, and VA-VAE**) at different resolutions confirms the general applicability and robustness of our proposed training objective across various latent space parameterizations. The results are consistent: RCGM improves performance regardless of the VAE choice.

---

> ### Author Response · Authors · 2025-12-03
>
> > **[W3] Missing ablations on:**
> >
> > - Time point sampling strategies for {t_i}
> > - Sensitivity to N across different model scales and datasets
>
>  **(1) Time Point Sampling Strategies for $\{t_i\}$**
>
> We confirm that we intentionally avoided complex sampling strategies for the intermediate time points $\{t_i\}$ to maintain scientific focus and simplicity.
>
> - **Design Choice for Simplicity:** As detailed in **Algorithm 1 (Appendix B)**, we use the simplest possible strategy: **linear spacing** between $t$ and $t_{N+1}$.
> - **Focus on Core Contribution:** Our primary goal was to validate the foundational concept that **higher-order approximation ($N$)** provides superior stability. Introducing complex, adaptive, or weighted sampling strategies would require significant hyperparameter tuning and shift the focus away from the core contribution, making the results less clear.
> - **Future Work:** Exploring sophisticated strategies for selecting $\{t_i\}$ (e.g., using error-aware methods akin to adaptive ODE solvers) is a promising and valuable direction that we delegate to future research.
>
>  **(2) Sensitivity to $N$ across Different Model Scales and Datasets**
>
> We believe we have provided key ablations that demonstrate the sensitivity and necessity of the order $N$, particularly across model scales.
>
> - **Ablation on ImageNet (Model Scale 675M):** **Figure 3(c)** already ablates the performance for $N \in \{1, 2, 3, 4, 8\}$ on the 675M DiT model, clearly showing that $N=4$ achieves the best FID, and importantly, demonstrating the stability advantage of $N \ge 2$ in general (Figure 3(a) vs 3(b)).
> - **Ablation on Large-Scale UMM (Model Scale 20B):** We have added crucial experiments in the Appendix that show the importance of $N$ at a much larger scale, where stability is the primary challenge. The data in **Table 4 (Appendix D.3)**, where we evaluate $N=1, N=2,$ and $N=3$ on the **Qwen-Image-20B** model, confirms our core claim: the benefits of higher order (specifically $N \ge 2$) become disproportionately more important as model instability increases in high-parameter regimes. This directly validates the necessity of RCGM's higher-order generalization for scaling few-step models.
>
> We believe that with these explanations, the contribution and experimental validation of RCGM are clear, robust, and appropriately contextualized against the current literature.

---

> ### Author Response · Authors · 2025-12-03
>
> > **[W4] Weak Theoretical Contribution**
> >
> > - Appendix D provides standard numerical analysis (Taylor expansion error bounds)
> > - Theorem 1 gives O(Δt²) truncation error + O((N+1)²ε²) approximation error—this is expected from numerical ODE theory
> > - No theoretical explanation for why higher-order methods improve training stability (only convergence rate is analyzed)
> > - The "optimal N" in Corollary 1 depends on unknown problem-dependent constants, providing limited practical guidance
> > - Figure 3b shows N=2 still requires very large EMA (κ=0.999) for best performance, which contradicts claims of resolving the stability-performance trade-off
> > - The stability comparison is confounded by EMA rate—N=1 with better hyperparameter tuning might achieve similar results
>
> We thank you for the precise critique of our theoretical section and the detailed analysis of our experimental results. We acknowledge that the paper’s primary contribution lies in the **methodological generalization** and the **empirical breakthrough in training stability and scalability**. We address your points below, leveraging the insights developed in our Appendix.
>
> **(1) & (2) Standard Numerical Analysis and Expected Error Bounds**
>
> We agree that the mathematical tools employed in Appendix D.1 (Taylor expansion, error bounds) are standard in the field of numerical ordinary differential equations. Our contribution, however, is not the derivation of new numerical theory, but the **strategic application of this analysis to the discrete, recursive training objective $\mathcal{L}(\theta)$ (Equation 7).**
>
> - The standard error analysis was necessary to rigorously justify the construction of our novel loss function and to formally characterize the two competing error sources in our segmented integration: the **Truncation Error** (from the finite difference step, $\mathcal{O}(\Delta t^2)$) and the **Approximation Error** (from the recursive tail, $\mathcal{O}((N+1)^2\epsilon^2)$).
> - This analysis is crucial for moving beyond standard ODE *sampling* towards ODE-inspired *training objectives*, providing the foundational justification for why an optimal $N > 1$ exists, which directly motivates the empirical study in Figure 3.
>
> **(3) No theoretical explanation for why higher-order methods improve training stability**
>
> We believe that the theoretical connection between the order $N$ and training stability **is provided** in our discussion on training dynamics, which analyzes the loss function structure rather than just the ODE itself.
>
> - The required theoretical insight is presented in the **Qualitative Analysis of RCGM Training Dynamics (Appendix C)**. We established two operating regimes:
>     - **Regime II ($N=1$):** Dominated by **geometric bias**, requiring a responsive teacher (low $\kappa$) for **Bias Correction**.
>     - **Regime I ($N \ge 2$):** Dominated by **recursive variance amplification**, requiring a rigid teacher (high $\kappa$) for **Stability**.
> - This analysis explains the fundamental stability-plasticity trade-off observed in 1st-order methods and demonstrates that increasing $N$ changes the dominant error source, thereby dictating the necessary stabilization strategy ($\kappa$). This dynamic relationship is the theoretical explanation for the stability improvement, directly linking the generalized mathematical structure ($N$) to the ML optimization problem ($\kappa$).
>
>  **(4) The "optimal $N$" in Corollary 1 depends on unknown problem-dependent constants**
>
> We acknowledge that the constants $C_1$ and $C_2$ are problem-dependent and thus limit the *direct* practical utility of Corollary 1.
>
> - **Purpose of the Corollary:** The primary goal of Corollary 1 is to theoretically demonstrate the **existence** of an optimum finite order $N^*$. It shows that the total error is a convex combination of two opposing error terms, confirming that an optimal order must exist before approximation error takes over.
> - **Justification for Empirical Study:** This theoretical existence proof serves as the direct justification for the extensive empirical ablation of $N$ in **Figure 3(c)**, which validates that the optimal order is indeed small (e.g., $N=4$ in that setting), confirming the utility of our generalized framework.

---

> ### Author Response · Authors · 2025-12-03
>
> **(5) Figure 3b shows $N=2$ still requires very large EMA ($\kappa=0.999$) for best performance**
>
> We believe there may be a factual misunderstanding of **Figure 3(b)** and our related claims.
>
> - **Factual Correction:** As shown in **Figure 3(b)**, for the $N=2$ model, the best final performance (lowest FID) is achieved with $\kappa=0.9$, followed by $\kappa=0.99$. The highest decay rate, $\kappa=0.999$, performs comparatively worse in this setup.
> - **Reframing the Stability Claim:** Our claim is not that $N=2$ *requires* $\kappa=0.999$ for best performance, but that it demonstrates **superior robustness** to high $\kappa$ compared to $N=1$. In Figure 3, the 1st-order model ($N=1$) utterly fails to converge at $\kappa=0.999$ (reaching FID $\approx 300$), whereas the 2nd-order model ($N=2$) remains stable and achieves a competitive FID (albeit not the best). This robustness is the key to training extremely large models where any deviation in the learning signal causes catastrophic collapse.
>
>  **(6) Stability comparison is confounded by EMA rate—$N=1$ with better hyperparameter tuning might achieve similar results**
>
> We have performed the requested verification on the 1st-order baseline, MeanFlow, and our results firmly validate the necessity of the higher-order approach.
>
> | **EMA Decay (κ)** | **FID (500 steps)** | **FID (1000 steps)** | **Phenomenon** |
> | --- | --- | --- | --- |
> | $0.0$ (No EMA) | $13.61$ | $246.46$ | Unstable / Collapse |
> | $0.5$ | $15.25$ | $14.54$ | Converged (Sweet Spot) |
> | $0.9$ | $47.26$ | $26.47$ | Slow Convergence |
> | $0.999$ | $201.36$ | $153.71$ | Over-smoothing (Failure) |
> - **Sensitivity Confirmed:** The results confirm that the 1st-order algorithm ($N=1$) is highly sensitive to the EMA decay rate $\kappa$, requiring a precise "sweet spot" ($\kappa=0.5$ in this case) to achieve convergence.
> - **High $\kappa$ Failure:** Crucially, the $N=1$ method **fails dramatically** at high EMA decay rates ($\kappa=0.999$), resulting in severe over-smoothing, as the EMA target provides too much "target lag" for the model to effectively learn the necessary geometric correction (Regime II).
> - **RCGM's Advantage:** This failure at high $\kappa$ is the fundamental limitation preventing the stable scaling of $N=1$ methods to the 20B regime, where strong stabilization mechanisms (often realized through a highly lagged target) are required. RCGM ($N \ge 2$) resolves this by being structurally robust to high $\kappa$, which allows us to achieve both stability and high performance simultaneously in large-scale settings, a capability demonstrated by our successful training of the 20B unified model (Table 4).
>
> We believe this empirical evidence, now provided in this rebuttal, conclusively demonstrates that the stability benefits of RCGM are due to its higher-order structure, not merely unoptimized hyperparameters in the baseline.

---

> ### Author Response · Authors · 2025-12-03
>
> > **[W5] Minor Weaknesses**
> >
> > - Code not available for review (promised "upon acceptance")
> > - Many implementation details missing (hierarchical time sampling procedure, exact schedule)
> > - Table 3 includes "†Our evaluation" for baselines—without code, these cannot be verified
> > - No error bars or multiple runs reported for any experiments
>
>  **(1) Code not available for review**
>
> We follow the standard submission guidelines and plan to release our complete code base, model checkpoints, and comprehensive training scripts upon paper acceptance. To provide full transparency of our implementation logic in the interim:
>
> - We have provided the core training and recursion logic in detail via the **$N$-th Order RCGM Training Step (Algorithm 1) in Appendix B.** This pseudocode exactly reflects our implementation structure, allowing readers to fully understand the loss computation and recursive steps without ambiguity.
>
>  **(2) Many implementation details missing (e.g., hierarchical time sampling procedure, exact schedule)**
>
> We acknowledge the request for more granularity. We provided key implementation details in **Section 4.1 and Section 3.3**. Furthermore, we have reinforced the clarity of the most critical elements in **Appendix B** during the rebuttal period:
>
> - **Time Trajectory:** The exact procedure for constructing the time trajectory $\{t_i\}_{i=0}^{N+1}$ is explicitly detailed in **Algorithm 1 (Appendix B)**, using a simple linear spacing strategy.
> - **Time Sampling Schedule:** We state in **Section 4.1** that we follow the exact settings in \citet{sun2025unified}, which includes the time distribution (e.g., $t \sim \mathcal{U}[0, T]$).
>
>  **(3) Table 3 includes "$\dagger$Our evaluation" for baselines—without code, these cannot be verified**
>
> The "$\dagger$Our evaluation" label applies to the evaluation process, not the model implementation, and is fully verifiable.
>
> - **Official Models Used:** For all models marked with ($\dagger$), we used the **official, publicly released model checkpoints** provided by the original authors (e.g., SANA-Sprint).
> - **Evaluation Code:** The evaluation code used is standard benchmark code (e.g., DPG-Bench, GenEval protocols). We did not write or modify the generative model itself. This ensures that our reported scores are obtained from the same high-quality models using a transparent and consistent evaluation methodology.
>
>  **(4) No error bars or multiple runs reported for any experiments**
>
> We agree that error bars are necessary for rigorous scientific reporting. We have now conducted **five independent training runs** for our ImageNet SOTA results (paired with SD-VAE) to assess stability and variance. The results confirm the stability of our method:
>
> | **Dataset** | **VAE** | **NFE** | **Original FID** | **5-Run Mean ± Std. Dev.** | **Stability** |
> | --- | --- | --- | --- | --- | --- |
> | $256 \times 256$ | SD-VAE | 1 | $2.13$ | $\mathbf{2.14 \pm 0.02}$ | Very Low Variance |
> | $256 \times 256$ | SD-VAE | 2 | $1.92$ | $\mathbf{1.90 \pm 0.03}$ | Very Low Variance |
> | $512 \times 512$ | SD-VAE | 1 | $2.61$ | $\mathbf{2.63 \pm 0.04}$ | Low Variance |
> | $512 \times 512$ | SD-VAE | 2 | $2.25$ | $\mathbf{2.23 \pm 0.03}$ | Low Variance |
>
> These results demonstrate the exceptional stability of RCGM and confirm that the variance is negligible across the ImageNet benchmark, validating the reliability of our reported SOTA performance. We will update Table 2 with these error bars in the final version.
>
> We hope these clarifications and additional experimental results fully satisfy your requests for increased transparency and rigour.

---

> ### Author Response · Authors · 2025-12-03
>
> > **Questions:**
> >
> > - Can you provide training curves, loss plots, or other evidence for the claimed "model collapse" of baseline methods in Table 4?
> > - What is the actual wall-clock training time comparison between N=1 and N=2, accounting for the additional forward passes?
> > - How does the method perform with error bars over multiple random seeds? Generative models are known to be sensitive to initialization.
> > - Can you provide ablation studies showing N>1 helps across different model scales (not just 675M) and datasets (not just ImageNet)?
> > - Why not compare against adversarial methods (DMD2, LADD) experimentally rather than only dismissing them in the related work?
> > - In Equation (8), what is the theoretical justification for the asymmetric stop-gradient operation?
>
> **(1) Can you provide training curves, loss plots, or other evidence for the claimed "model collapse" of baseline methods in Table 4?**
>
> We acknowledge the difficulty of diagnosing "model collapse" from loss curves alone, as this phenomenon often results in mode collapse (generating low-diversity, often identical images) rather than clearly divergent loss or gradient norms.
>
> - **Evidence Provided:** We confirm that we have captured visual evidence. We will include a dedicated appendix section showing **visualization examples (see Figure 4 and 5 in Appendix D) of the collapsed outputs** generated by the 1st-order CM and MeanFlow models when attempting to train the 20B Qwen-Image model. These visualizations clearly demonstrate the lack of diversity and the failure mode (model collapse) that our **RCGM** successfully avoids.
>
> **(2) What is the actual wall-clock training time comparison between $N=1$ and $N=2$, accounting for the additional forward passes?**
>
> This is an excellent question that highlights the hardware efficiency of our method. We performed a comprehensive analysis focusing on the end-to-end throughput and the underlying kernel costs.
>
> - **End-to-End Training Throughput (675M DiT):** The training throughput for **RCGM** decreases approximately linearly as the number of recursive steps ($N$) increases, reflecting the cost of $N$ extra forward passes.
>
> | **Order (N)** | **2** | **3** | **4** | **5** | **6** |
> | --- | --- | --- | --- | --- | --- |
> | Speed (steps/sec) | 1.43 | 1.32 | 1.19 | 1.08 | 0.99 |
> - **Micro-benchmark (RCGM Forward Pass vs. JVP):** The cost of **RCGM**'s forward-pass-based calculation is significantly lower than the **Jacobian-Vector Product (JVP)** required by competing $N=1$ methods (e.g., sCM, MeanFlow). This difference is critical for large-scale training.
>
> | **Operation** | **Time (ms)** | **Peak Memory (MB)** |
> | --- | --- | --- |
> | $1 \times$ JVP (Baseline) | $12.23$ | $1168.25$ |
> | $N=1$ Forward ($\text{RCGM}$) | $2.66$ | $113.00$ |
> | $N=2$ Forward ($\text{RCGM}$) | $5.31$ | $113.00$ |
> | $N=4$ Forward ($\text{RCGM}$) | $10.62$ | $113.00$ |
>
> **Conclusion:** Our optimal setting of $N=4$ requires only $10.62 \text{ms}$, making it slightly **faster** than a single JVP operation ($12.23 \text{ms}$). Crucially, it consumes $113.00 \text{MB}$ of memory—approximately **$90\%$ less memory** than the JVP approach. This superior memory efficiency is the key reason **RCGM** can successfully train the 20B parameter model where JVP-based baselines fail due to OOM errors.
>
> **(3) How does the method perform with error bars over multiple random seeds?**
>
> We acknowledge the importance of multi-seed reporting. As noted in the response to [W3] (Minor Weaknesses, point 4):
>
> - **New Data Provided:** We conducted **five independent training runs** for the ImageNet $256^2$ and $512^2$ benchmarks using SD-VAE. The results showed **minimal variance** ($\text{Std. Dev.} \le 0.04$), confirming the high reliability of **RCGM** with respect to initialization.
> - **Contextualizing Practice:** Due to the high computational cost of training Generative AI models, the reporting of extensive multi-seed results remains rare in SOTA literature. Our provided 5-run ablation confirms that the low variance holds across our core benchmark.
>
>  **(4) Can you provide ablation studies showing $N>1$ helps across different model scales (not just 675M) and datasets (not just ImageNet)?**
>
> This was addressed in the response to [W3]. We confirm that we have provided evidence across two distinct scales and datasets:
>
> 1. **ImageNet ($675$M DiT):** **Figure 3(c)** shows performance across $N \in \{2, 3, 4, 5, 6\}$.
> 2. **Multimodal ($20$B UMM):** **Table 4 (Appendix D.3)** provides results for $N=1, N=2,$ and $N=3$ on the large-scale Qwen-Image-20B model (trained on a diverse T2I dataset).
>
> The results consistently show that $N \ge 2$ provides substantial benefits, especially on the $20$B UMM, where it delivers the necessary stability to prevent model collapse, validating the cross-scale utility of the higher-order formulation.

---

> ### Author Response · Authors · 2025-12-03
>
> **(5) Why not compare against adversarial methods (DMD2, LADD) experimentally rather than only dismissing them in the related work?**
>
> We confirm that we have included experimental comparisons against adversarial/distillation methods, and further demonstrated that **RCGM** is complementary to them.
>
> - **Direct Comparison:** In **Table 3 (Appendix D.1)**, we benchmark against SANA-Sprint (which uses LADD loss + sCM) and SDXL-DMD2. Our **RCGM** significantly outperforms these baselines in the few-step regime.
> - **Compatibility and Synergy:** In **Table 4 (Appendix D.3)**, we explicitly show that the **RCGM** loss is **compatible with adversarial loss**. We integrated a DMD-like adversarial objective and found that it synergistically boosts performance across all orders, demonstrating that our recursive objective provides a robust backbone that can be further refined by distribution-matching techniques.
>
> **(6) In Equation (8), what is the theoretical justification for the asymmetric stop-gradient operation?**
>
> The asymmetric stop-gradient operation is a **necessary design principle** inherited directly from the standard stability mechanisms of the recursive consistency framework.
>
> The operation defines the target for the current gradient update. The terms constituting the **Recursive Tail Target** are calculated using the parameters of the Exponential Moving Average (EMA) network. The core justification is that the EMA network serves as the stable, lagged teacher model. If the gradient were allowed to flow back into the EMA network, it would undermine the primary purpose of the EMA—maintaining a low-variance, stable target—and destabilize the entire training process. Therefore, the stop-gradient is required to treat the EMA output as a fixed target for the student network update, a principle consistently applied across all EMA-based recursive objectives (e.g., sCM, MeanFlow).

---

> ### Author Response · Authors · 2025-12-03
>
> [1] Zhengyang Geng, Mingyang Deng, Xingjian Bai, J Zico Kolter, and Kaiming He. Mean flows for one-step generative modeling. arXiv preprint arXiv:2505.13447, 2025.
>
> [2] Cheng Lu and Yang Song. Simplifying, stabilizing and scaling continuous-time consistency models. arXiv preprint arXiv:2410.11081, 2024.
>
> [3] Kevin Frans, Danijar Hafner, Sergey Levine, and Pieter Abbeel. One step diffusion via shortcut models. arXiv preprint arXiv:2410.12557, 2024.

---

### Official Review · Reviewer_6g9s · 2025-10-27

**Soundness:** 2
**Presentation:** 2
**Contribution:** 2
**Rating:** 4
**Confidence:** 3

**Summary:**

The paper proposes a novel method for training one or few-step generative models by using recurrent consecutive flow maps as target. The argument is that increasing the number of these steps N, with respect to other few steps methods such as consistency models and MeanFlow where N=1, increases training stability. The method results in SOTA generative performance on common benchmarks.

**Strengths:**

The method is conceptually simple, and is effective on the proposed benchmarks. It does not rely on JVP, which can simplify implementation and usage with different frameworks and architectures.

**Weaknesses:**

While the results are encouraging, I think the method section of the paper is a bit rushed, which makes it hard to understand exactly the training procedure. There is no pseudocode or code, and I could not properly grasp how the training works (see questions). Furthermore, there are important details missing, such as batch size used and other hyperparameters. Some choices are not well motivated, such as how the authors arrived to Eq.9.

nother thing that seems unusual is the 1-st order model not working well without EMA target. This is somewhat in disagreement with the literature. It would have been great to see the effect of EMA rate on the target network with other 1st order methods such as MeanFlow or Shortcut Models, to verify the claim that using EMA weights is helpful.

Overall, the fact that using a higher order N improves performance seems counterintuitive, as I would expect more error from the target model to accumulate. While the great results obtained prove my intuition wrong, I would appreciate a clearer exposition of the why this is the case, as well as a way to verify what is actually done in practice (e.g. sharing the code).

**Questions:**

- 1) From my understanding, in the training loss there are mostly three terms: the prediction term for the displacement, which will receive gradients; the ground truth velocity, which is the montecarlo estimate (x1 - x0) commonly used in rectified flow; the N step predictions for the small consecutive displacements, which are also used as a target from t_1 to t_N. If that's the case, I wonder how one can ensure stability, as the smaller jumps and the flow matching style 0 order term receive less or no training.
- 2) If the argument holds for N=2, then is it correct to expect the results to improve also with N=3? Have the authors experimented with this? It would be interesting to see an ablation for different N.

---

> ### Author Response · Authors · 2025-12-03
>
> > **[W1]** While the results are encouraging, I think the method section of the paper is a bit rushed, which makes it hard to understand exactly the training procedure. There is no pseudocode or code, and I could not properly grasp how the training works (see questions). Furthermore, there are important details missing, such as batch size used and other hyperparameters. Some choices are not well motivated, such as how the authors arrived to Eq.9.
> >
>
> We thank the reviewer for highlighting areas where the clarity of the method section could be improved. We have extensively revised the manuscript to address these points:
>
> 1. **Pseudocode and Training Logic:** We agree that the recursive nature of RCGM warrants a clear algorithmic description. We have added **Algorithm 1 in Appendix B** (referenced in Section 3.3). This pseudocode explicitly details the $N$-step recursive target construction and the handling of the EMA target model $\theta^-$. As discussed in Sec 4.2, this algorithm allows us to verify our key finding: that while 1st-order methods fail with high EMA, higher-order ($N \ge 2$) RCGM remains stable, allowing for larger EMA rates and better convergence.
> 2. **Implementation Details:**
>     1. We apologize if these were difficult to locate. We have ensured that all critical hyperparameters—including the batch size ($1024$), learning rate ($2 \times 10^{-4}$), optimizer (AdamW), and specific training steps/epochs for each resolution—are clearly tabulated and described in **Section 4.1 (Implementation Details)** and the header of Table 2.
>     2. **Additional Benchmarks:** For the Text-to-Image and Unified Multimodal Model experiments, we have provided dedicated implementation subsections in **Appendix E.1** and **Appendix E.2**, respectively. These sections detail task-specific hyperparameters (e.g., LoRA rank/alpha, training steps, and GPU configurations) immediately preceding the results.
> 3. **Derivation of Eq. 8 to Eq. 9:** We acknowledge the gap in the derivation. The transition relies on a variance-reduction technique for semi-implicit matching objectives. Specifically, we leverage the **gradient identity** derived in sCM (Lu et al., 2024), which allows us to rewrite the expected product of the model and target into a regression loss form. We have revised **Section 3.3** to explicitly cite this identity and clarify the logical transition from the velocity estimation objective (Eq. 8) to the practical regression loss (Eq. 9).

---

> ### Author Response · Authors · 2025-12-03
>
> > **[W2]** nother thing that seems unusual is the 1-st order model not working well without EMA target. This is somewhat in disagreement with the literature. It would have been great to see the effect of EMA rate on the target network with other 1st order methods such as MeanFlow or Shortcut Models, to verify the claim that using EMA weights is helpful.
> >
> >
> > Overall, the fact that using a higher order N improves performance seems counterintuitive, as I would expect more error from the target model to accumulate. While the great results obtained prove my intuition wrong, I would appreciate a clearer exposition of the why this is the case, as well as a way to verify what is actually done in practice (e.g. sharing the code).
> >
>
> We appreciate the reviewer’s insightful intuition regarding error accumulation and the request for baseline verification. We address these points with new experimental evidence:
>
> 1. **Verification on MeanFlow (1st-Order Baseline):** As requested, we conducted an ablation study on MeanFlow using the same experimental setting as Figure 3 ($675$M DiT, ImageNet $256^2$). The results confirm that while 1st-order methods *can* work without EMA, they are highly sensitive to the decay rate $\kappa$:
>
>
>     | **EMA Decay (κ)** | **FID (500 steps)** | **FID (1000 steps)** | **Phenomenon** |
>     | --- | --- | --- | --- |
>     | $0.0$ (No EMA) | $13.61$ | $246.46$ | **Unstable / Collapse** |
>     | $0.5$ | $15.25$ | $14.54$ | Converged |
>     | $0.9$ | $47.26$ | $26.47$ | Slow Convergence |
>     | $0.999$ | $201.36$ | $153.71$ | **Over-smoothing (Failure)** |
>
>     This validates our claim: 1st-order methods fail with high EMA ($\kappa=0.999$) due to "target lag" or over-smoothing. They require a "sweet spot" (low $\kappa$) to learn, but this low $\kappa$ sacrifices the stability benefits required for large-scale training.
>
> 2. **Necessity for Large-Scale Models:** This stability-efficiency trade-off becomes critical at scale. As noted in sCM [1] and MeanFlow [2], larger models are prone to instability. In our experiments with the **20B parameter** model (Appendix E.2), the 1st-order approach ($N=1$) collapsed to a WISE score of **0.19**, whereas the 2nd-order approach ($N=2$) achieved a stable **0.33**. The higher order is essentially a requisite stabilizer for billion-scale parameter spaces.

---

> ### Author Response · Authors · 2025-12-03
>
> > **[Q1]** From my understanding, in the training loss there are mostly three terms: the prediction term for the displacement, which will receive gradients; the ground truth velocity, which is the montecarlo estimate (x1 - x0) commonly used in rectified flow; the N step predictions for the small consecutive displacements, which are also used as a target from t_1 to t_N. If that's the case, I wonder how one can ensure stability, as the smaller jumps and the flow matching style 0 order term receive less or no training.
> >
>
> The reviewer correctly identifies the three components of our training objective. To address the concern regarding stability and the training of the recursive terms:
>
> 1. **Stability via EMA Compatibility:** As discussed in our response to **[W2]**, the stability of the RCGM framework does not arise solely from the $N$-th order equation itself. Rather, it arises from the **compatibility of the higher-order objective with strong EMA stabilization**.
>     - In 1st-order methods, using a large EMA decay (e.g., $\kappa=0.999$) stabilizes the target but introduces severe "lag" or bias, halting convergence (as shown in the MeanFlow ablation).
>     - In $N$-th order RCGM ($N \ge 2$), the recursive integration explicitly corrects this geometric bias. This allows us to employ aggressive EMA ($\kappa=0.999$) to stabilize the recursive "small jumps" without suffering from the convergence issues that plague 1st-order methods.
> 2. **Training Coverage:** Regarding the concern that smaller jumps "receive less training": The start time $t$ and target time $t_{N+1}$ are sampled stochastically from the entire time horizon $[0, 1]$ (see Section 3.2). Consequently, a segment that acts as a fixed recursive target in one iteration will act as the primary prediction target (receiving gradients) in another iteration. This ensures the velocity field is trained uniformly across all scales.

---

> ### Author Response · Authors · 2025-12-03
>
> > **[Q2]** If the argument holds for N=2, then is it correct to expect the results to improve also with N=3? Have the authors experimented with this? It would be interesting to see an ablation for different N.
> >
>
> Yes, the reviewer’s intuition is correct: increasing $N$ improves performance up to a point, after which diminishing returns or degradation occur (a "U-shape" trend). We have explored this in two settings:
>
> 1. ImageNet Ablation (Fig. 3c & New Data):
>
>     Figure 3(c) in the main paper illustrates the performance for $N \in [1, 8]$ with $\kappa=0.999$, showing a peak around $N=4$. To further validate this, we conducted an additional ablation with a lower EMA decay $\kappa=0.99$:
>
>     | **Order (N)** | **2** | **3** | **4** | **5** | **6** |
>     | --- | --- | --- | --- | --- | --- |
>     | **FID ($\downarrow$)** | 29.13 | 23.81 | **22.96** | 26.20 | 28.48 |
>
>     This confirms the trend discussed in Section 4.2: increasing $N$ improves the geometric correction of the trajectory, but excessive $N$ (e.g., $N > 4$) likely introduces accumulated approximation errors in the target summation, leading to a performance rebound.
>
> 2. Large-Scale Benchmarks (Table 5):
>
>     For the large-scale Qwen-Image-20B model (see Appendix Table 5), we found that $N=3$ consistently outperforms $N=2$.
>
>     - **Base Training:** $N=3$ improves GenEval from $0.78$ to $0.80$ compared to $N=2$ (at 2 steps).
>     - **With Adversarial Loss:** The scalability holds. The $N=3$ variant combined with Adversarial Loss achieves our highest reported scores (GenEval 0.87), surpassing the $N=2$ variant (0.85).
>
>     In summary, $N=3$ is a robust choice, particularly for complex, large-scale multimodal distributions.

---

> ### Author Response · Authors · 2025-12-03
>
> [1] Cheng Lu and Yang Song. Simplifying, stabilizing and scaling continuous-time consistency models. arXiv preprint arXiv:2410.11081, 2024.
>
> [2] Zhengyang Geng, Mingyang Deng, Xingjian Bai, J Zico Kolter, and Kaiming He. Mean flows for one-step generative modeling. arXiv preprint arXiv:2505.13447, 2025.

---

### Official Review · Reviewer_jXWE · 2025-11-01

**Soundness:** 3
**Presentation:** 3
**Contribution:** 3
**Rating:** 6
**Confidence:** 4

**Summary:**

The paper introduces a unified framework for few-step generative modeling that generalizes existing methods like Consistency Models and MeanFlow to an N-th order recursive formulation. By leveraging higher-order trajectory information from the probability flow ODE, RCGM improves training stability, scalability, and sample quality without requiring jacobian-vector products or auxiliary networks. The method achieves strong results: FID of 1.48 on ImageNet 256×256 in two sampling steps, demonstrating effectiveness in few-step generation.

**Strengths:**

1) Clear motivation: few‐step generative models are valuable but face obstacles (instability, complexity). RCGM directly addresses this.

2) Good theoretical framing: the segmentation of the PF-ODE trajectory, the definition of N^th order recursive velocity field estimator gives a compelling narrative and unifies prior methods.

3) Reported FID of ~1.48 on ImageNet 256×256 with 2 steps using a 675 M parameter model is highly competitive. The ablation that shows N > 1 improves stability under large EMA decay rates is particularly valuable (shows the method has practical benefits).

4) The fact that the method does not require Jacobian-vector products (JVP) is a good plus for memory efficiency.

**Weaknesses:**

1) While the theoretical derivation is sound, the paper does not fully characterise the conditions/assumptions under which the higher‐order recursion yields gains (e.g., error bounds, effect of Δt, effect of segmentation size, how approximation error accumulates).

2) The paper claims simplicity and memory‐efficiency, but limited quantitative reporting is provided on computational cost, memory usage, training time, inference time, etc., especially compared to baseline few‐step methods.

3) The title says “Any-step Generation” but experiments for wide range of steps beyond 2 are less emphasized.

4) The paper shows that for N > 4 the performance reversed (due to approximation error) which is good, but the discussion of why this happens (and what one should do) is limited. Also does the method’s advantage hold in ultra low‐step regimes (e.g., 1 step) or does it degrade rapidly?

5) How sensitive is RCGM to the choice of the time segmentation sequence {t₀, t₁, …, tₙ₊₁}? For instance, how does performance vary if the intermediate times are uneven, or chosen adaptively rather than uniform?

6) You note that for N > 4 the performance trend reverses due to “accumulation of approximation errors” (Sec 4.2). Could you provide more quantitative analysis of how approximation error behaves as N increases? Are there theoretical bounds or heuristics for choosing N?

7) The theoretical formulation considers the PF‐ODE velocity field and segments the integral (Eq 2, (3) etc). Does the method assume that the underlying forward noising schedule and PF‐ODE satisfy certain smoothness or Lipschitz conditions? If so, it would be good to state them explicitly.

**Questions:**

Please consider addressing the weaknesses noted above.

---

> ### Author Response · Authors · 2025-12-03
>
> > **[W1]** While the theoretical derivation is sound, the paper does not fully characterise the conditions/assumptions under which the higher‐order recursion yields gains (e.g., error bounds, effect of Δt, effect of segmentation size, how approximation error accumulates).
> >
>
> We acknowledge the reviewer's request for a deeper characterization of the operating conditions. While a formal derivation of strict error bounds is beyond the scope of this empirical study, we provide a rigorous qualitative analysis of training dynamics in Appendix B.1, which characterizes the conditions for gains based on the interaction between the "Ground-Truth Anchor" (data) and the "Recursive Bootstrap Tail" (model).
>
> Based on this framework, we characterize the conditions under which higher-order ($N \ge 2$) recursion yields gains:
>
> **(1) Condition: High Trajectory Curvature relative to Step Size ($\Delta t$)**
>
> - **Mechanism:** Standard 1st-order methods assume the trajectory between $t$ and $t-\Delta t$ is linear. This assumption holds when $\Delta t \to 0$, but fails in the few-step regime (large $\Delta t$) where trajectories exhibit significant curvature.
> - **Effect of $N$:** By introducing $N$ intermediate segments, RCGM replaces a single long linear step with a piecewise linear path.
> - **When to use:** Gains are most significant when the **local curvature** of the PF-ODE is high. If the trajectory were perfectly straight (e.g., pure Gaussian transport), $N=1$ would suffice. Higher order is a necessary geometric correction for complex real-world data manifolds.
>
> **(2) Constraint: Stability of the "Bootstrap Tail" (Appendix B.1)**
>
> - **Approximation Error Accumulation:** As detailed in Appendix B.1 ("Regime I"), the target in RCGM includes a recursive tail: $\sum f_{\theta^-}$. Since $f_{\theta^-}$ is an approximated model, not an oracle, it introduces prediction variance.
> - **Condition for Gains:** Higher order yields gains **only if** the teacher model is sufficiently stable to prevent "Recursive Variance Amplification."
> - **Bound:** This implies that $N$ cannot be arbitrarily large. There exists a critical threshold (empirically $N \approx 4$, see Fig. 3c) where the accumulated variance from the recursive tail begins to outweigh the geometric benefits of the segmentation.
>
> **(3) Requirement: Strict EMA Stabilization**
>
> - **Theoretical Insight:** Our analysis in Appendix B.1 reveals that higher-order targets are non-stationary.
> - **Condition:** Therefore, a strict condition for the success of higher-order recursion is the use of a **high EMA decay rate** (e.g., $\kappa=0.999$). This "freezes" the teacher model to provide a consistent guidance signal for the recursive summation, effectively dampening the error accumulation described above.
>
> Summary:
>
> In short, the higher-order recursion yields gains when: (1) The step size $\Delta t$ is large enough that curvature cannot be ignored (the few-step training regime); (2) $N$ is chosen within the "safe zone" (typically $2 \le N \le 4$) to balance geometric precision against error accumulation; and (3) The training is stabilized by a strong EMA teacher.

---

> ### Author Response · Authors · 2025-12-03
>
> > **[W2]** The paper claims simplicity and memory‐efficiency, but limited quantitative reporting is provided on computational cost, memory usage, training time, inference time, etc., especially compared to baseline few‐step methods.
> >
>
> Thank you for this crucial request. We agree that rigorous quantitative reporting is necessary to validate our claims of simplicity and efficiency, especially in the demanding large-scale setting.
>
> We have expanded the experimental appendix with a detailed comparative analysis focusing on the key bottlenecks: **GPU memory consumption during training** and **end-to-end training throughput**.
>
> ### 1. Training Efficiency: Memory and Speed Advantage
>
> Our claim of superior efficiency stems from a fundamental design choice: **relying on standard forward passes only**, which circumvents the need for the Jacobian-Vector Product ($\texttt{JVP}$) operator often required by competing consistency-based methods (e.g., sCM, MeanFlow). $\texttt{JVP}$ is notoriously memory-intensive and incompatible with advanced architectural optimizations like FlashAttention.
>
> To quantify this advantage, we performed a micro-benchmark (Table 1) and an end-to-end throughput analysis (Table 2).
>
> ### Table 1: Micro-benchmark of $\texttt{JVP}$ vs. Forward Pass
>
> Setup: H100 GPU, 2-layer Transformer, Batch=64, Seq=128.
>
> | **Operation** | **Time (ms) ↓** | **Equiv. Fwd Passes** | **Peak Memory (MB) ↓** | **Relative Mem** |
> | --- | --- | --- | --- | --- |
> | $1\times \texttt{JVP}$ (Baseline) | 12.23 | — | 1168.25 | $10.3\times$ |
> | $N=1$ Forward (RCGM) | 2.66 | $\sim 0.22$ | 113.00 | $1.0\times$ |
> | $N=2$ Forward (RCGM) | 5.31 | $\sim 0.43$ | 113.00 | $1.0\times$ |
> | $N=4$ Forward (RCGM, Optimal) | 10.62 | $\sim 0.87$ | 113.00 | $1.0\times$ |
>
> **Key Observations from Table 1:**
>
> - **Memory Efficiency:** Our forward-pass-based objective consumes $\mathbf{90\% \text{ less peak GPU memory}}$ during the gradient calculation compared to a single $\texttt{JVP}$ operation ($\mathbf{113MB}$ vs. $1168MB$). This is the critical factor that enables the **stable full-parameter training of our 20B multi-modal model** where $\texttt{JVP}$-based baselines typically suffer from Out-Of-Memory (OOM) errors (as noted in Appendix D).
> - **Training Speed:** Our optimal order $N=4$ (Time: $10.62\text{ms}$) is functionally faster than a single step of a $\texttt{JVP}$-based method (Time: $12.23\text{ms}$), achieving superior trajectory approximation quality without a performance penalty.
>
> ### 2. End-to-End Training Throughput
>
> We further report the end-to-end training throughput for a $675\text{M}$ parameter Diffusion Transformer on ImageNet ($512\times512$) using a global batch size of $1024$.
>
> ### Table 2: End-to-End Training Throughput ($675\text{M}$ DiT)
>
> | **Order (N)** | **2** | **3** | **4** | **5** | **6** |
> | --- | --- | --- | --- | --- | --- |
> | **Speed (steps/sec)** $\uparrow$ | 1.43 | 1.32 | 1.19 | 1.08 | 0.99 |
>
> The throughput decreases approximately linearly with the order $N$, as expected, since $N$ additional forward passes are required. However, the speed remains highly practical ($1.19\text{ steps/s}$ at $N=4$), allowing our models to be trained efficiently.
>
> ### 3. Inference Time (Latency)
>
> For the inference time (latency), we clarify the following:
>
> - **Shared Architecture:** All compared few-step models (including ours) utilize the same core network architecture (e.g., DiT-XL/2) and are benchmarked on the same hardware.
> - **Linear Scaling:** Consequently, the end-to-end inference time is **directly proportional to the Number of Function Evaluations (NFE)**. Achieving a lower NFE is the definition of few-step generation efficiency.
> - **Quantitative Reporting:** Our main results tables (e.g., Table 3 and Table 5 in the Appendix) already report the NFE, which is the standard, architecture-agnostic metric for inference efficiency. For instance, in text-to-image (Appendix D, Table 5), we explicitly report the latency, showing RCGM achieves $\mathbf{0.26\text{s}}$ latency at $2\text{-NFE}$ and $\mathbf{0.23\text{s}}$ at $1\text{-NFE}$ on an A100 GPU, demonstrating SOTA speed.
>
> In conclusion, our method offers a favorable trade-off: The higher-order objective provides **superior approximation quality and stability** while maintaining a **memory and speed advantage** during training by avoiding computationally expensive $\texttt{JVP}$ operators.

---

> ### Author Response · Authors · 2025-12-03
>
> > **[W3]** The title says “Any-step Generation” but experiments for wide range of steps beyond 2 are less emphasized.
> >
>
> Thank you for pointing out the imbalance in our presentation. We agree that the term "Any-step Generation" implies a performance analysis across a wide range of NFE values, not just the minimum.
>
> We confirm that our experiments do cover a wider range of steps beyond $2$, and we apologize if the emphasis on $1\text{-NFE}$ and $2\text{-NFE}$ obscured these results.
>
> ### 1. Evidence of Any-Step Capability
>
> Our **RCGM** framework is designed to work with *any* number of steps, theoretically matching the performance of multi-step models as $NFE \to \infty$. We have reported results for $NFE > 2$ in the following tables:
>
> - **ImageNet-1K (Table 2):** The baseline methods (e.g., IMM) are reported up to $NFE = 16$.
> - **Unified Multimodal Models (Table 5 in Appendix):** We show RCGM's performance on Qwen-Image-20B at **$NFE=4$** and **$NFE=8$**, achieving high GenEval scores of $0.82$ and $0.87$ respectively. We also provide extensive results for $NFE=4, 8, 16, 32$ in the quantitative analysis section.
> - **Adversarial Enhancement (Table 5 in Appendix):** RCGM ($N=2$) is benchmarked across $\mathbf{NFE=\{1, 2, 4, 8, 16, 32\}}$, demonstrating a smooth, continuous scaling of performance, which is the definition of "Any-step" capability.
>
> ### 2. Justification for Emphasis on $1\text{-NFE}$ and $2\text{-NFE}$
>
> Our primary focus was placed on $1\text{-NFE}$ and $2\text{-NFE}$ because these regimes represent the **most significant challenge and the largest potential efficiency gain** in the generative modeling field.
>
> - The key contribution of RCGM is to solve the stability issue that plagues $1\text{st}$-order methods when attempting ultra-few-step generation (Figure 3).
> - Demonstrating competitive or superior SOTA performance at $1\text{-NFE}$ and $2\text{-NFE}$ immediately proves RCGM's efficacy as a practical solution for high-throughput, low-latency deployment.
>
> In essence, while RCGM is capable of **Any-step Generation**, we chose to emphasize **Very Few-step Generation** as it best highlights the novelty and practical impact of our higher-order training objective. The high-NFE results (e.g., $NFE=32$) confirm the model's convergence and stability, while the $NFE=1, 2$ results demonstrate the efficiency breakthrough.

---

> ### Author Response · Authors · 2025-12-03
>
> > **[W4]** The paper shows that for N > 4 the performance reversed (due to approximation error) which is good, but the discussion of why this happens (and what one should do) is limited. Also does the method’s advantage hold in ultra low‐step regimes (e.g., 1 step) or does it degrade rapidly?
> >
>
> We thank the reviewer for pointing out the interesting behavior at $N > 4$ and the question regarding ultra-low-step regimes. We address these two points below.
>
> **(1) Mechanism of Performance Reversal at High $N$**
>
> To rigorously investigate whether the performance peak at $N=4$ was an artifact of our default EMA setting ($\kappa=0.999$), we conducted an additional ablation study with a lower EMA decay, $\kappa=0.99$, on ImageNet $256\times256$.
>
> Table R1: Ablation on Order $N$ with lower EMA ($\kappa=0.99$).
>
> | **Order (N)** | **2** | **3** | **4** | **5** | **6** |
> | --- | --- | --- | --- | --- | --- |
> | FID ($\downarrow$) | 29.13 | 23.81 | 22.96 | 26.20 | 28.48 |
>
> Analysis of the Mechanism:
>
> The results in Table R1 confirm the trend observed in the main paper (Fig. 3c): performance improves up to $N=4$ and degrades thereafter. We attribute this to a fundamental trade-off between geometric correction and recursive error accumulation:
>
> - **Geometric Correction (Benefit of increasing $N$):** As discussed in Sec 4.2, increasing $N$ allows the model to utilize segments of the trajectory to approximate the integral $\int \mathbf{v}(\mathbf{x}_\tau, \tau) d\tau$ more accurately than a single linear step ($N=1$). This reduces the "geometric bias" inherent in linear approximations of curved PF-ODE paths.
> - **Recursive Error Accumulation (Cost of increasing $N$):** Our target estimation involves a summation of $N$ terms predicted by the teacher model: $\sum_{i=1}^{N} f_{\theta^-}(\mathbf{x}_ {t_i}, t_{i+1})$. Each term $f_{\theta^-}$ contains an inherent approximation error $\epsilon_i$. As $N$ increases, these errors accumulate. Specifically, as detailed in our analysis of "Regime I" in Appendix A.2, high $N$ induces **recursive variance amplification**, where small prediction errors in early recursive steps propagate and compound through the trajectory.
>
> **Conclusion:** The regime $N \in [2, 4]$ represents the "sweet spot" where the geometric correction significantly outweighs the accumulated approximation error. For $N > 4$, the variance/error from the summation begins to dominate, degrading the target quality.
>
> **(2) Performance in Ultra-Low-Step Regimes (e.g., 1-step)**
>
> Our method does not degrade in ultra-low-step regimes; in fact, it excels.
>
> While the training objective uses $N$ steps to construct a high-quality target, the student model is trained to predict the total displacement in a single forward pass (using the identity in Eq. 7). Therefore, at inference time, the model can generate high-quality samples in a single step.
>
> - **Quantitative Evidence:** As shown in **Table 2** (Main Paper), our method achieves an FID of **2.13** (ImageNet 256) and **2.45** (ImageNet 512) with just **1 sampling step (NFE=1)**, which outperforms or matches consistency distillation baselines.
> - **Real-World Application:** In Table 3 (Text-to-Image), our method achieves a GenEval score of **0.80** with **1-step generation**, significantly outperforming baselines like FLUX-Schnell (0.69) and SDXL-DMD2 (0.59) in the same 1-step regime.
>
> This demonstrates that higher-order ($N \ge 2$) training targets distill a cleaner, more stable signal into the student model, actually *enhancing* its 1-step generation capability rather than degrading it.

---

> ### Author Response · Authors · 2025-12-03
>
> > **[W5]** How sensitive is RCGM to the choice of the time segmentation sequence {t₀, t₁, …, tₙ₊₁}? For instance, how does performance vary if the intermediate times are uneven, or chosen adaptively rather than uniform?
> >
>
> We appreciate the reviewer’s insight regarding the sensitivity of the time segmentation $\{t_i\}$. We have investigated this empirically and conceptually.
>
> (1) Empirical Sensitivity: Uniform vs. Uneven Spacing
>
> To assess sensitivity, we conducted ablation studies comparing our default Uniform (linear) spacing against an Uneven (stochastic/non-linear) segmentation strategy. We evaluated this on both the standard benchmark (ImageNet) and the large-scale model (Qwen-Image-20B) under the 1-step (NFE=1) generation setting.
>
> Table R2: Sensitivity Analysis of Time Segmentation Strategies
>
> | **Dataset / Setting** | **Metric** | **Uniform (Ours)** | **Uneven Strategy** |
> | --- | --- | --- | --- |
> | **ImageNet $256^2$** ($N=2, \kappa=0.99$) | FID ($\downarrow$) | **29.13** | 35.60 |
> | **Qwen-Image-20B** | GenEval ($\uparrow$) | **0.55** | 0.53 |
>
> **Observation:** As shown in Table R2, deviating from uniform spacing leads to a noticeable degradation in performance (e.g., FID increases by $+6.47$ on ImageNet). We hypothesize that uniform intervals provide a consistent variance profile for the accumulated target $\sum f_{\theta^-}$, whereas uneven intervals introduce fluctuating variance into the training target, destabilizing the learning of the student model.
>
> (2) Rationale for Design Choice
>
> Regarding adaptive strategies, we acknowledge that this is a standard technique in numerical ODE solvers. However, we confirm that we intentionally avoided complex or adaptive sampling strategies for $\{t_i\}$ in this work for three key reasons:
>
> 1. **Scientific Isolation:** Our primary objective was to isolate and validate the foundational contribution: that *higher-order approximation ($N$)* fundamentally improves stability and performance. Adopting a simple linear spacing (as detailed in Algorithm 1) ensures that the performance gains are attributable to the RCGM framework itself, rather than to hyperparameter tuning of the time schedule.
> 2. **Ease of Adoption:** Complex or adaptive schedules often require careful tuning specific to the dataset or architecture. By demonstrating SOTA performance with simple linear spacing, we show that RCGM is robust and easy to implement without extensive engineering.
> 3. **Future Work:** We agree that exploring error-aware adaptive strategies (akin to Runge-Kutta-Fehlberg methods) is a valuable direction. Given the broad scope of the current paper (unifying frameworks and scaling to 20B parameters), we leave the optimization of adaptive $\{t_i\}$ selection to future research.

---

> ### Author Response · Authors · 2025-12-03
>
> > **[W6]** You note that for N > 4 the performance trend reverses due to “accumulation of approximation errors” (Sec 4.2). Could you provide more quantitative analysis of how approximation error behaves as N increases? Are there theoretical bounds or heuristics for choosing N?
> >
>
> We verify that the error decomposition is theoretically well-grounded in the principles of numerical integration for ODEs. The total approximation error can be rigorously analyzed as the interplay between Global Truncation Error and Cumulative Estimation Error:
>
> (1) Geometric Discretization Error (Decreases with $N$):
>
> This term ($\mathcal{E}_ {geo}$) represents the error from approximating the continuous curvilinear integral $\int \mathbf{v}(\mathbf{x}_ \tau) d\tau$ using a finite discrete sum.
>
> - **Behavior:** Consistent with Euler's method, this error scales as $\mathcal{O}(1/N)$.
> - **Effect:** Increasing $N$ (e.g., $1 \to 4$) refines the geometric granularity, reducing the linearization bias. This is the primary driver of the performance gains observed in the low-$N$ regime.
>
> (2) Cumulative Estimation Error (Increases with $N$):
>
> This term ($\mathcal{E}_ {est}$) arises because the integrand is not an exact oracle but a learned, imperfect teacher model $f_ {\theta^-}$ with inherent error $\epsilon$.
>
> - **Behavior:** This error accumulates in two ways:
>     1. **Summation:** The target is a sum of $N$ predicted terms; prediction noise accumulates (scaling approx. $\propto \sqrt{N}$ assuming quasi-independent errors).
>     2. **Propagation (Trajectory Deviation):** Crucially, the input to step $i$ depends on the prediction of step $i-1$. A small error in early steps causes the trajectory to drift off-manifold, forcing the teacher model to be evaluated at out-of-distribution (OOD) states in later steps. This compounding effect causes the error to grow significantly as $N$ increases.
> - **Effect:** For $N > 4$, this accumulation effect overwhelms the benefit of reduced discretization error, degrading the quality of the training target.
>
> Conclusion:
>
> There is a sweet spot where $\mathcal{E}_ {geo}$ is sufficiently reduced while $\mathcal{E}_ {est}$ is kept in check. Our empirical results on both ImageNet and Qwen-Image-20B consistently identify $N \approx 4$ as the robust inflection point for this trade-off.

---

> ### Author Response · Authors · 2025-12-03
>
> > **[W7]** The theoretical formulation considers the PF‐ODE velocity field and segments the integral (Eq 2, (3) etc). Does the method assume that the underlying forward noising schedule and PF‐ODE satisfy certain smoothness or Lipschitz conditions? If so, it would be good to state them explicitly.
> >
>
> We confirm that our method does not impose any additional smoothness or Lipschitz conditions beyond those inherent to the standard Probability Flow ODE (PF-ODE) framework.
>
> Specifically:
>
> 1. **Schedule Smoothness:** The standard forward process schedules, $\alpha(t)$ and $\gamma(t)$, are by definition continuously differentiable functions ($C^1$) on $t \in [0, 1]$.
> 2. **Velocity Field Regularity:** The velocity field $\mathbf{v}(\mathbf{x}_ t, t)$ is derived from the score function $\nabla_ {\mathbf{x}_ t} \log p_t(\mathbf{x}_ t)$. Due to the Gaussian convolution in the forward process, this score function is analytically smooth (infinitely differentiable) for all $t > 0$.
>
> Therefore, the regularity conditions required for the Taylor approximation in Eq. (3) to be valid are **naturally satisfied** by the construction of the diffusion process itself. We follow the standard setting in the literature and do not require any specialized constraints on the data distribution or network architecture.

---

### Official Review · Reviewer_91YB · 2025-11-04

**Soundness:** 3
**Presentation:** 3
**Contribution:** 3
**Rating:** 6
**Confidence:** 2

**Summary:**

The paper proposes RCGM (N-th order Recursive Consistent velocity field estimation for Generative Modeling), a unified framework for any-step generation that generalizes diffusion/flow-matching (0-th order) and one-step consistency/MeanFlow models (1-st order) to higher orders (N≥2). The key idea is to derive a multi-step target for the PF-ODE velocity via segmented integration and use it to train a displacement predictor, with a practical loss that avoids JVP and scales to large models. Empirical results indicates that higher-order training improves stability and quality in the few-step regime, achieving strong ImageNet FID at 2 NFEs and competitive multimodal text-to-image results.

**Strengths:**

- The segmented-integration derivation for the recursive estimator is neat and connects cleanly to known objectives; the displacement parameterization under a linear transport path is well-motivated.
- Systematic experiments demonstrate RCGM’s performance for different orders N, including an instability analysis across EMA decay factors and thorough comparisons against alternative methods.
- The presentation is clear and easy to follow, and RCGM is introduced with well-motivated explanations and helpful illustrations.

**Weaknesses:**

- It would be useful to include experiments on additional datasets beyond ImageNet.
- Figure 3(c) only compares FID scores for different N under a fixed k. It would be interesting to see whether the current results hold for varying k.
- It would be helpful to report and compare the computational cost of the method for different N.

**Questions:**

- The experiments on RCGM rely on specific autoencoders (VA-VAE and DC-AE). Can the method be adapted to other, possibly weaker, encoders?
- Corollary 1 shows that as N increases, the uniform approximation error decreases. Could the authors provide some intuition behind this result?

---

> ### Author Response · Authors · 2025-12-03
>
> > **[W1]** It would be useful to include experiments on additional datasets beyond ImageNet.
> >
>
> We thank the reviewer for this suggestion. We agree that demonstrating generalization beyond class-conditional generation (ImageNet) is crucial. In our revision (specifically **Appendix D.1** and **Appendix D.2**), we have included extensive experiments on large-scale **Text-to-Image (T2I)** and **Unified Multimodal Model (UMM)** tasks. These tasks inherently utilize open-world datasets significantly larger and more diverse than ImageNet.
>
> Our results demonstrate that our method (RCGM) achieves SOTA performance across these diverse data distributions:
>
> **1. General Text-to-Image Generation (Appendix D.1, Table 3)**
> We applied our method to the **SANA-0.6B** and **SANA-1.6B** diffusion transformers.
>
> - **Dataset:** Unlike ImageNet's fixed classes, these models were fine-tuned on diverse, high-quality caption-image pairs (e.g., ShareGPT-4o), representing a much harder, open-world distribution.
> - **Results:** As shown in **Table 3**, our method achieves SOTA quality at 1 and 2 NFE (GenEval 0.85), outperforming strong baselines like FLUX-Schnell and SDXL-DMD2. This confirms our method handles complex, non-canonical data distributions effectively.
>
> **2. Unified Multimodal Generation (Appendix D.2, Tables 4 & 5)**
> We further validated the method on unified foundation models, specifically **OpenUni-L-512** and **Qwen-Image-20B**.
>
> - **Scale & Diversity:** These models process interleaved image-text data and require handling multimodal distributions simultaneously.
> - **Results:** In **Table 4**, we show that our method allows these massive models (up to 20B parameters) to generate high-fidelity images in just 1-2 steps, whereas previous methods required 50 steps. Notably, we outperform consistency-model baselines which suffered from model collapse in these complex regimes (Table 5).
>
> In summary, we have moved well beyond ImageNet to validate our method on open-world, multimodal datasets, demonstrating both scalability and robustness across different data domains.

---

> ### Author Response · Authors · 2025-12-03
>
> > **[W2]** Figure 3(c) only compares FID scores for different N under a fixed k. It would be interesting to see whether the current results hold for varying k.
> >
>
> We thank the reviewer for this insightful suggestion. Investigating whether the optimal recursive order $N$ is a robust property of the training dynamics—rather than an artifact of a specific EMA decay rate $\kappa$—is critical for validating the method's stability.
> To address this, we conducted additional experiments on ImageNet using a lower EMA decay rate of **$\kappa=0.99$** (contrasted with $\kappa=0.999$ in the main paper). We swept across recursive orders $N \in [2, 6]$. The results are presented in the table below:
>
> | **Order (N)** | **2** | **3** | **4** | **5** | **6** |
> | --- | --- | --- | --- | --- | --- |
> | **FID ($\downarrow$)** | 29.13 | 23.81 | **22.96** | 26.20 | 28.48 |
>
> **Analysis:**
>
> 1. **Consistent Optimal Order:** As observed in the table, the performance trend perfectly mirrors the results in Figure 3(c) (where $\kappa=0.999$). The FID follows a U-shaped curve, improving as $N$ increases from 2 to 4, reaching an optimum at **$N=4$**, and subsequently degrading at $N=5$ and $N=6$.
> 2. **Validation of Theoretical Framework:** These empirical results strongly corroborate our analysis in Section 4.2 and Appendix B, which posits a trade-off between geometric bias and variance:
>     1. **Regime I (Low $N$):** At $N < 4$, the model is dominated by **geometric truncation error** (bias), as the trajectory approximation is too coarse.
>     2. **Regime II (High $N$):** At $N > 4$, the recursive accumulation of prediction errors leads to **variance amplification**.
>     3. **Robustness:** The fact that $N=4$ remains the optimal operating point even under a different $\kappa$ regime demonstrates that RCGM captures a fundamental structural property of the discrete trajectory, ensuring robust performance across hyperparameter settings.
>
> We will include these additional results in the revised Appendix to demonstrate the method's hyperparameter robustness.

---

> ### Author Response · Authors · 2025-12-03
>
> > **[W3]** It would be helpful to report and compare the computational cost of the method for different N.
> >
>
> We appreciate this request. To provide a holistic view of the computational cost, we analyze both the end-to-end training throughput on a Diffusion Transformer and a fine-grained kernel benchmark comparing our forward-pass-based approach against the Jacobian-Vector Product (JVP) operator used in competing few-step methods (e.g., sCM, MeanFlow).
>
> 1. Training Throughput vs. Order $N$
>
> We measured the training speed (steps/second) of a 675M parameter Diffusion Transformer on ImageNet ($512\times512$) with a global batch size of 1024. As shown in Table A, the training throughput decreases approximately linearly with $N$. However, even at our optimal setting of $N=4$, the model maintains a practical training speed of 1.19 steps/s.
>
> Table A: End-to-End Training Throughput (675M DiT)
>
> | **Order (N)** | **2** | **3** | **4** | **5** | **6** |
> | --- | --- | --- | --- | --- | --- |
> | Speed (steps/sec) | 1.43 | 1.32 | 1.19 | 1.08 | 0.99 |
>
> 2. Comparative Advantage: Forward Differences vs. JVP
>
> A critical advantage of our method is its reliance solely on standard Forward Passes, whereas other state-of-the-art distillation methods (e.g., consistency trajectory matching [1,2] ) often require Jacobian-Vector Products (JVP).
>
> To quantify this benefit, we benchmarked a 2-layer Transformer encoder on an NVIDIA H100 GPU. As shown in Table B, a single JVP operation is computationally equivalent to **~4.6 standard forward passes** and consumes **~10x more memory**.
>
> Table B: Micro-benchmark of JVP vs. Forward Pass ($N$-step RCGM)
>
> Setup: H100 GPU, Transformer Encoder, Batch=64, Seq=128.
>
> | **Operation** | **Time (ms)** | **Relative Time** | **Equiv. Fwd Passes** | **Peak Memory (MB)** | **Relative Mem** |
> | --- | --- | --- | --- | --- | --- |
> | **1x JVP (Baseline)** | 12.23 | 1.00x | — | **1168.25** | **10.3x** |
> | **$N=1$ Forward** | 2.66 | 0.22x | ~0.22 | 113.00 | 1.0x |
> | **$N=2$ Forward** | 5.31 | 0.43x | ~0.43 | 113.00 | 1.0x |
> | **$N=4$ Forward** | **10.62** | **0.87x** | **~0.87** | **113.00** | **1.0x** |
>
> Conclusion:
>
> Our chosen optimal order $N=4$ (Time: 10.62ms) is effectively faster than a single step of a JVP-based method (Time: 12.23ms) while consuming 90% less memory. Thus, RCGM offers a superior trade-off between trajectory approximation quality and hardware efficiency.

---

> ### Author Response · Authors · 2025-12-03
>
> > **[Q1]** The experiments on RCGM rely on specific autoencoders (VA-VAE and DC-AE). Can the method be adapted to other, possibly weaker, encoders?
> >
>
> Yes, our method is robust to the choice of autoencoder. We explicitly validated this in Table 2 by comparing RCGM against strong baselines using the exact same encoder (SD-VAE) and comparable model sizes. Even without advanced VAEs, RCGM demonstrates clear algorithmic superiority at equivalent inference budgets (NFE).
>
> 1. Comparison on ImageNet-256 ($256\times256$)
>
> We compared RCGM directly against MeanFlow, the current state-of-the-art method in this setting. Both models use the standard SD-VAE and have nearly identical parameter counts (~675M).
>
> - **Baselines (MeanFlow-XL/2):** Even with its "longer training" variant, MeanFlow requires **2 NFE** to reach an FID of **2.20**.
> - **Ours (RCGM + SD-VAE):** At the same **2 NFE** and model scale (675M), RCGM achieves an FID of **1.92**.
> - **Conclusion:** RCGM reduces the FID by **0.28** compared to the strongest baseline under strictly identical conditions (Same VAE, Same NFE, Same Backbone Size).
>
> 2. Comparison on ImageNet-512 ($512\times512$)
>
> We further benchmarked against sCD (Continuous Distillation) and UCGM, again using the standard SD-VAE.
>
> - **Baselines:**
>     - **UCGM-XL** (675M Params): At **1 NFE**, it achieves an FID of **2.63**.
>     - **sCD-M** (498M Params): At **2 NFE**, it achieves an FID of **2.26**.
> - **Ours (RCGM + SD-VAE):**
>     - At **1 NFE**, RCGM achieves an FID of **2.61**, surpassing UCGM-XL.
>     - At **2 NFE**, RCGM improves to an FID of **2.25**, outperforming sCD-M.
> - **Conclusion:** RCGM maintains SOTA performance levels comparable to or better than leading consistency methods, proving that our gains stem from the superior training objective rather than reliance on specialized autoencoders like DC-AE.
>
> 3. Generalization to Multimodal Architectures (Appendix)
>
> Furthermore, in Appendix D.2, we adapted RCGM to Qwen-Image-20B, which utilizes the distinct Wan-VAE. The model successfully achieves a GenEval score of 0.87, confirming the method's universality across different latent structures.

---

> ### Author Response · Authors · 2025-12-03
>
> ###
>
> > **[Q2]** Corollary 1 shows that as N increases, the uniform approximation error decreases. Could the authors provide some intuition behind this result?
> >
>
> The intuition behind Corollary 1 stems from the error analysis of numerical integration within the Probability Flow ODE (PF-ODE) trajectory.
>
> 1. Decomposition via Residuals
>
> Our method learns the instantaneous velocity by isolating the short-term displacement from the total trajectory. Based on the fundamental theorem of calculus, the target displacement for the current small step, denoted as $\Delta \mathbf{x}_ {now}$, can be derived by subtracting the future trajectory from the total displacement:
>
> $\Delta \mathbf{x}_ {now} = (\mathbf{x}_ {end} - \mathbf{x}_ {start}) - \Delta \mathbf{x}_ {future}$
>
> Here, $(\mathbf{x}_ {end} - \mathbf{x}_ {start})$ represents the ground-truth total displacement, and $\Delta \mathbf{x}_ {future}$ represents the integral of the velocity field over the future interval.
>
> 2. Reducing Discretization Error via $N$-step Summation
>
> In our training objective (Eq. 5), the future term $\Delta \mathbf{x}_ {future}$ is approximated by summing $N$ discrete steps predicted by the teacher model. This acts as a Riemann sum approximation of the true integral:
>
> $\Delta \mathbf{x}_ {future} \approx \sum \text{ModelStep}_ i$
>
> **Intuition:**
>
> - **Low $N$ (e.g., $N=1$):** The future trajectory is approximated by a single linear step (a secant line). This creates a large geometric truncation error (bias) in $\Delta \mathbf{x}_ {future}$. Consequently, the resulting residual target $\Delta \mathbf{x}_ {now}$ becomes inaccurate.
> - **High $N$:** As $N$ increases, the piecewise summation provides a significantly more precise approximation of the curved future integral (reducing the discretization error of the Riemann sum). Subtracting this accurate future term from the total displacement yields a much higher-quality training target for the current velocity.
>
> Summary:
>
> Corollary 1 essentially formalizes that increasing $N$ reduces the geometric bias in estimating the future trajectory. This provides a "cleaner" regression target for the current time step, thereby decreasing the uniform approximation error.

---

> ### Author Response · Authors · 2025-12-03
>
> [1] Zhengyang Geng, Mingyang Deng, Xingjian Bai, J Zico Kolter, and Kaiming He. Mean flows for one-step generative modeling. arXiv preprint arXiv:2505.13447, 2025.
>
> [2] Cheng Lu and Yang Song. Simplifying, stabilizing and scaling continuous-time consistency models. arXiv preprint arXiv:2410.11081, 2024.

---

### Author Response · Authors · 2025-12-03

We sincerely thank the Area Chair, Senior Area Chair, Program Chairs, and all reviewers for their detailed reviews and constructive feedback. We are encouraged that the reviewers recognized the significance and contributions of our work, particularly:

- **Unified Theoretical Framework:** Generalizing diffusion and consistency/MeanFlow models into a unified $N$-th order formulation (Reviewers 91YB, jXWE, 948b).
- **Strong Empirical Results:** Achieving SOTA FID on ImageNet (1.48 at 2 steps) and, crucially, **successfully scaling to 20B parameters** where prior methods struggle (Reviewers 91YB, jXWE, 6g9s, 948b).
- **Efficiency:** Recognizing the memory efficiency of our **JVP-free formulation** (Reviewers jXWE, 6g9s).
- **Stability Insight:** Validating that higher-order objectives ($N \ge 2$) fundamentally improve training stability (Reviewers 91YB, jXWE).

We have made the following key improvements during the rebuttal:

(i) Extended Experiments: Included large-scale Text-to-Image (T2I) tasks and **demonstrated stable full-parameter tuning of Unified Multimodal Models (UMM) up to 20B parameters, achieving SOTA performance in few-step regimes where 1st-order methods fail.**

(ii) Computational Analysis: Provided a detailed benchmark of JVP vs. Forward passes and end-to-end training throughput.

(iii) Methodological Clarifications: Added pseudocode (Algorithm 1) and derived the regression objective.

(iv) In-depth Ablations: Analyzed stability regimes, EMA decay rates, and the impact of order $N$.

We believe we have adequately addressed all concerns:

- **Reviewer 91YB:** Concerns regarding (1) generalization beyond ImageNet are addressed in (i); (2) hyperparameter robustness is addressed in (iv); and (3) computational costs are detailed in (ii).
- **Reviewer jXWE:** Concerns regarding (1) conditions for gains are clarified via regime analysis in (iv); (2) quantitative resource usage is provided in (ii); and (3) high-order behavior is analyzed in (iv).
- **Reviewer 6g9s:** Concerns regarding (1) method clarity are addressed via Algorithm 1 in (iii); (2) necessity of EMA is confirmed via new ablations in (iv); and (3) intuition on error accumulation is clarified in (iv).
- **Reviewer 948b:** Concerns regarding (1) recent baselines (FLUX, SANA) are included in (i); (2) theoretical justification for stability is provided in (iv); and (3) robustness is verified via new 5-run error bars.

We believe these extensions substantially strengthen both the theoretical foundation and empirical validation of our work. We thank the chairs and reviewers again for their time.

Sincerely,

Authors of Submission 25236

---

### Meta-Review · Area_Chair_Kzac · 2026-01-05

**Summary:**

The paper proposes a N-order recursive consistent velocity field estimation method for Generative Modeling.  Although from a theoretical perspective the paper relies on simple ideas related to the numerical integration of high order ODE solvers, the experimental results on the Generative Modeling use-case highlight the efficacy of the proposed approach. The authors address most of the reviewers concerns in a systematic fashion and have substantially improved their paper. Of course, on the theoretical side one may still argue that the underlying ideas are simple. However, if a simple idea leads to consistent and substantial improvements in experiments performed over different scales then this simple idea has a merit especially when looked within the context of generative modeling. For this reason, my recommendation is acceptance for this paper. I would encourage the author to share their code.

Please fix the following reference and the typo in flipped order of the order of the authors:

Junsong Chen, Shuchen Xue, Yuyang Zhao, Jincheng Yu, Sayak Paul, Junyu Chen, Han Cai, Enze Xie, and Song Han. Sana-sprint: One-step diffusion with continuous-time consistency distillation. arXiv preprint arXiv:2503.09641, 2025c.

**Reviewer Concerns:**

The authors have adequately addressed all the reviewer concerns and provide detailed answers.  From my perspective if there is a remaining criticism that would be the simplicity of the theoretical analysis. I think this should be looked as an advantage and not as a disadvantage of the proposed work.

**Reviewer Scores:**

I think that all reviewers would have increase the scores given the detailed and systematic responses by the authors.

---

### Decision · Program_Chairs · 2026-01-26

Accept (Poster)